# BEYOND HEURISTICS: GLOBALLY OPTIMAL CONFIGURATION OF IMPLICIT NEURAL REPRESENTATIONS

## ABSTRACT

Implicit Neural Representations (INRs) have emerged as a transformative paradigm in signal processing and computer vision, excelling in tasks from image reconstruction to 3D shape modeling. Yet their effectiveness is fundamentally limited by the absence of principled strategies for optimal configuration—spanning activation selection, initialization scales, layer-wise adaptation, and their intricate interdependencies. These choices dictate performance, stability, and generalization, but current practice relies on ad-hoc heuristics, brute-force grid searches, or task-specific tuning, often leading to inconsistent results across modalities. This work introduces OptiINR, the first unified framework that formulates INR configuration as a rigorous optimization problem. Leveraging Bayesian optimization, OptiINR efficiently explores the joint space of discrete activation families—such as sinusoidal (SIREN), wavelet-based (WIRE), and variable-periodic (FINER)—and their associated continuous initialization parameters. This systematic approach replaces fragmented manual tuning with a coherent, data-driven optimization process. By delivering globally optimal configurations, OptiINR establishes a principled foundation for INR design, consistently maximizing performance across diverse signal processing applications.

## 1 INTRODUCTION

Implicit Neural Representations (INRs), also referred to as coordinate-MLPs, have fundamentally reshaped how continuous signals are represented and processed across domains from computer vision to computational physics (Li et al., 2021; Xie et al., 2022). In contrast to traditional discrete representations tied to fixed spatial resolutions, INRs parameterize signals as continuous functions via neural networks, yielding resolution-independent representations with exceptional expressiveness and memory efficiency. This paradigm has unlocked capabilities that were previously unattainable, powering applications such as Neural Radiance Fields (NeRF) for photorealistic view synthesis Mildenhall et al. (2020), signed distance functions for high-fidelity 3D reconstruction (Park et al., 2019; Mescheder et al., 2019), advanced medical imaging, and even neural solvers for partial differential equations (Sitzmann et al., 2020a; Raissi et al., 2019). The strength of INRs lies in their ability to exploit the universal approximation property of neural networks (Cybenko, 1989; Hornik et al., 1989) to learn complex, high-dimensional mappings from coordinate space to signal values. Landmark works such as DeepSDF (Park et al., 2019) demonstrated that MLPs can learn continuous signed distance functions for representing 3D geometry, while NeRF showed that similar architectures can capture view-dependent radiance fields with high fidelity (Mildenhall et al., 2020). Together, these advances established INRs as a powerful alternative to grid-based representations.

Despite substantial progress, the practical effectiveness of implicit neural representations (INRs) remains constrained by a *capacity–convergence gap* rooted in the tight coupling between activation families and their initialization schemes. High-capacity activations—sinusoidal (SIREN), wavelet-based/Gabor (WIRE), Gaussian, and variable-periodic (FINER) (Sitzmann et al., 2020a; Saragadam et al., 2023; Ramasinghe and Lucey, 2022; Liu et al., 2024) — provide rich spectral control but can be acutely sensitive to initialization; conversely, simpler, more stable choices converge reliably yet underfit high-frequency content. Initialization strategies (*e.g.,*, SIREN 's scale-preserving design) are therefore not interchangeable: the optimal settings depend on activation-specific properties, yielding a high-dimensional, non-convex search landscape where activation and initialization cannot be tuned independently. In practice, small hyperparameter changes can shift performance by over 10 dB PSNR on the same task, yet prevailing workflows still rely on manual, heuristic-driven tuning or coarse grid search. These observations underscore that bridging the capacity–convergence gap requires **joint,**

**principled optimization** of activation selection and initialization to achieve stable training, strong generalization, and robust performance.

To bridge the capacity-convergence gap and move beyond heuristic tuning, we introduce OptiINR (Optimal INR Configuration via Bayesian Optimization), a unified framework that recasts INR configuration as a formal global-optimization problem over a high-dimensional, mixed-variable space. Because each evaluation entails end-to-end training, we employ Bayesian optimization (Jones et al., 1998; Snoek et al., 2012) — designed for expensive black-box objectives — to navigate a comprehensive search space spanning activation families (*e.g.,* SIREN, WIRE, FINER, Gauss, FR) (Sitzmann et al., 2020a; Saragadam et al., 2023; Liu et al., 2024; Jayasundara et al., 2025; Ramasinghe and Lucey, 2022) and their conditional hyperparameters (*e.g.,* base frequency, spread/scale, initialization scaling) Tancik et al. (2021); Sitzmann et al. (2020a). Activation selection is modeled as categorical, while associated parameters are continuous and conditional on the chosen family, enabling sample-efficient exploration of the complex, non-linear performance landscape and discovery of high-performing, robust configurations for specific INR tasks. Unlike fragmented trial-and-error, OptiINR provides an automated, scientifically grounded procedure for configuring state-of-the-art INRs. Our contributions are:

- We introduce OptiINR, a Bayesian optimization framework that jointly optimizes activation families and their initialization parameters, replacing manual heuristic-driven tuning with principled, globally-aware configuration search. We provide theoretical justification in Section G.2 demonstrating convergence guarantees for our approach.

- We formalize INR configuration via a multilayer search space that integrates state-of-the-art activation families and initialization schemes under a single optimization formulation.

- Across canonical INR tasks — 1D audio reconstruction, 2D image representation, 3D shape prediction — OptiINR consistently discovers superior configurations and outperforms hand-tuned baselines under the same evaluation budgets.

- OptiINR yields robust configurations that mitigate the hypersensitivity of certain activations to initialization, broadening practical applicability across diverse signal modalities.

## 2 Background

**Implicit Neural Representations.** An Implicit Neural Representation (INR) parameterizes a continuous signal $g : \mathcal{X} \subset \mathbb{R}^d \to \mathcal{Y} \subset \mathbb{R}^m$ as a neural network $f_\theta$, typically an $L$-layer MLP (Sitzmann et al., 2019; Li et al., 2021), encoding the signal within its parameters $\theta$. This paradigm offers fundamental advantages over discrete representations: resolution independence and memory efficiency, as storage scales with network complexity rather than sampling density. The forward pass through the network is defined recursively: $\mathbf{z}^{(0)} = \gamma(\mathbf{x})$, $\mathbf{z}^{(\ell)} = \sigma_{\mathbf{p}}(\mathbf{W}^{(\ell)}\mathbf{z}^{(\ell-1)} + \mathbf{b}^{(\ell)})$ for $\ell = 1, \ldots, L-1$, and $f_\theta(\mathbf{x}) = \mathbf{W}^{(L)}\mathbf{z}^{(L-1)} + \mathbf{b}^{(L)}$, where $\theta = \{\mathbf{W}^{(\ell)}, \mathbf{b}^{(\ell)}\}_{\ell=1}^{L}$ are the learnable parameters with $\mathbf{W}^{(\ell)} \in \mathbb{R}^{h_\ell \times h_{\ell-1}}$ and $\mathbf{b}^{(\ell)} \in \mathbb{R}^{h_\ell}$, $\sigma_{\mathbf{p}}$ is an element-wise activation function with parameters $\mathbf{p}$, and $\gamma$ is an optional coordinate encoding. Given a dataset $\mathcal{D} = \{(\mathbf{x}_i, \mathbf{y}_i)\}_{i=1}^{N}$ sampled from the ground truth signal, we optimize $\theta^\star = \arg\min_\theta \frac{1}{N} \sum_{i=1}^{N} \ell(f_\theta(\mathbf{x}_i), \mathbf{y}_i) + \mathcal{R}(\theta)$, where $\ell$ is a task-specific loss function and $\mathcal{R}$ is an optional regularization term. While the Universal Approximation Theorem (Cybenko, 1989; Hornik et al., 1989) guarantees theoretical expressivity, a fundamental practical challenge is spectral bias (Rahaman et al., 2019; Canatar et al., 2021): neural networks trained with gradient descent inherently learn low-frequency components before high-frequency ones, yielding overly smooth reconstructions that fail to capture fine-grained details and sharp transients in natural signals. Consequently, the performance of $f_{\theta^\star}$ depends critically on architectural choices made prior to training—particularly the activation function family and parameter initialization strategy—which together determine optimization stability, frequency expressivity, and generalization capacity.

**Activation Functions and Spectral Bias** The evolution of activation functions in INR literature directly addresses the fundamental challenge of spectral bias. Initial attempts with standard activations like ReLU proved insufficient, necessitating positional encoding(Ramasinghe and Lucey, 2022) - a preprocessing step mapping input coordinates to higher-dimensional Fourier feature spaces to make high-frequency variations accessible. A conceptual breakthrough came with Sinusoidal Representation Networks (SIREN), which integrate periodicity directly into the network architecture by employing $\sigma_{\mathbf{p}}(x) = \sin(\omega_0 x)$ as the primary activation. SIREN demonstrated that appropriately chosen activations could obviate positional encoding; however, their success depends critically on principled initialization schemes that preserve activation distributions across layers, highlighting the tight coupling between activation choice and initialization strategy. Subsequent research questioned

the necessity of periodicity itself, producing a powerful toolkit of activation functions with distinct spectral properties. Gaussian activations, $\sigma_{\mathbf{p}}(x) = e^{-(s_0 x)^2}$, offer non-periodic alternatives with controllable spatial extent through scale parameter $s_0$. Wavelet Implicit Representations (WIRE) employ Gabor wavelets (Saragadam et al., 2023), $\sigma_{\mathbf{p}}(x) = e^{j\omega_0 x} e^{-|s_0 x|^2}$, valued for their optimal space-frequency concentration that minimizes the uncertainty principle—particularly suitable for visual signal representation. More recent frameworks like FINER and FINER++ (Liu et al., 2024) introduce variable-periodic functions, $\sigma_{\mathbf{p}}(x) = \sin(\omega_0(|x| + 1)x)$, which modulate local frequency based on input magnitude through adaptive bias initialization, enabling flexible spectral control across different signal regions. While this evolutionary path has produced increasingly sophisticated activation functions, each advancement introduces sensitive hyperparameters (e.g., $\omega_0, s_0, k$) requiring specific initialization strategies. This proliferation creates a complex configuration landscape where performance depends critically on joint optimization of activation family, parameter values, and initialization scheme—reinforcing the need for principled, automated configuration strategies.

**Automated Model Configuration**  The challenge of automatically configuring machine learning models is addressed by Automated Machine Learning (AutoML) and Neural Architecture Search (NAS) (Elsken et al., 2019; Feurer and Hutter, 2019). Our work, OptiINR, operates within this paradigm to find optimal hyperparameter configurations for single, specific tasks (e.g., representing a given image). This approach is distinct from, yet complementary to, meta-learning for INRs. Meta-learning approaches such as MetaSDF or Meta-SparseINR Sitzmann et al. (2020b) learn weight initializations from signal distributions, enabling rapid fine-tuning for unseen signals by optimizing network weights for fast adaptation across tasks. In contrast, OptiINR optimizes network hyperparameters (architecture) for maximal performance on individual target signals.

**Gaussian Processes**  A Gaussian Process (GP) is a non-parametric Bayesian model that defines a probability distribution over functions (Rasmussen and Williams, 2006), making it a powerful tool for regression tasks where the underlying function is unknown. A function $f$ drawn from a GP is denoted as $f(\mathbf{x}) \sim \mathcal{GP}(m(\mathbf{x}), k(\mathbf{x}, \mathbf{x}'))$, where $m(\mathbf{x}) = \mathbb{E}[f(\mathbf{x})]$ is the mean function and $k(\mathbf{x}, \mathbf{x}') = \mathbb{E}[(f(\mathbf{x}) - m(\mathbf{x}))(f(\mathbf{x}') - m(\mathbf{x}'))]$ is the covariance (kernel) function. The kernel is a symmetric, positive semi-definite function encoding prior beliefs about function properties such as smoothness and length-scale. For regression with observed data $\mathcal{D} = \{(\mathbf{x}_i, y_i)\}_{i=1}^n$, a GP infers a posterior distribution over functions. A key property is that any finite collection of function values is jointly Gaussian distributed. The posterior predictive distribution for a test point $\mathbf{x}_*$ is also Gaussian: $p(f(\mathbf{x}_*)|\mathcal{D}, \mathbf{x}_*) = \mathcal{N}(\mu(\mathbf{x}_*), \sigma^2(\mathbf{x}_*))$ with predictive mean $\mu(\mathbf{x}_*) = \mathbf{k}_*^T(\mathbf{K} + \sigma_n^2\mathbf{I})^{-1}\mathbf{y}$ and variance $\sigma^2(\mathbf{x}_*) = k(\mathbf{x}_*, \mathbf{x}_*) - \mathbf{k}_*^T(\mathbf{K} + \sigma_n^2\mathbf{I})^{-1}\mathbf{k}_*$, where $\mathbf{K}_{ij} = k(\mathbf{x}_i, \mathbf{x}_j)$ is the $n \times n$ kernel matrix, $\mathbf{k}_* = [k(\mathbf{x}_*, \mathbf{x}_1), \ldots, k(\mathbf{x}_*, \mathbf{x}_n)]^T$ is the vector of covariances between test and training points, $\mathbf{y}$ is the vector of observed outputs, and $\sigma_n^2$ is the observation noise variance. The predictive mean $\mu(\mathbf{x}_*)$ provides the best estimate of the function value, while the predictive variance $\sigma^2(\mathbf{x}_*)$ quantifies uncertainty—a property fundamental to the intelligent search strategy of Bayesian optimization.

# 3 Method

The performance of an Implicit Neural Representation is critically sensitive to its architectural configuration, particularly the layer-wise selection of activation functions and corresponding weight initialization schemes. This sensitivity creates a "capacity-convergence gap," where theoretically powerful architectures fail to realize their potential due to the difficulty of finding stable and effective configurations. Current practices rely on manual tuning, parameter reuse, or greedy layer-wise optimization, none of which guarantee global optimality. We propose a novel framework that recasts this complex, ad-hoc process as a formal global optimization problem, solved efficiently using Bayesian optimization to search the high-dimensional, mixed-variable space of network architectures. This principled approach automates the discovery of globally optimal configurations, moving beyond the limitations of existing methods.

**Bayesian Optimization for Expensive Black-Box Functions.**  Bayesian optimization is a sample-efficient methodology for global optimization of expensive-to-evaluate, black-box functions (Jones et al., 1998; Snoek et al., 2012; Shahriari et al., 2015). It is particularly well-suited for problems of the form $\boldsymbol{\lambda}^* = \arg\max_{\boldsymbol{\lambda} \in \Lambda} f(\boldsymbol{\lambda})$ where $f(\boldsymbol{\lambda})$ is an objective function with unknown analytic form and costly evaluation. The methodology comprises two primary components: a probabilistic surrogate model and an acquisition function. The surrogate model approximates the objective function probabilistically. We employ a Gaussian Process (GP), a non-parametric Bayesian regression model defining a prior distribution over functions: $f \sim \mathcal{GP}(m(\boldsymbol{\lambda}), k(\boldsymbol{\lambda}, \boldsymbol{\lambda}'))$, where $m(\boldsymbol{\lambda})$ is the

mean function and $k(\boldsymbol{\lambda}, \boldsymbol{\lambda}')$ is the covariance kernel. Given observations $\mathcal{D}_n = \{(\boldsymbol{\lambda}_i, y_i)\}_{i=1}^n$ where $y_i = f(\boldsymbol{\lambda}_i)$, the GP posterior provides a predictive distribution for any unevaluated point $\boldsymbol{\lambda}_*$: $p(f(\boldsymbol{\lambda}_*)|\mathcal{D}_n, \boldsymbol{\lambda}_*) = \mathcal{N}(\mu(\boldsymbol{\lambda}_*), \sigma^2(\boldsymbol{\lambda}_*))$. The predictive mean $\mu(\boldsymbol{\lambda}_*)$ estimates the function value, while variance $\sigma^2(\boldsymbol{\lambda}_*)$ quantifies uncertainty. An acquisition function $\alpha(\boldsymbol{\lambda})$ uses these statistics to balance exploration and exploitation, guiding the search for the next evaluation point: $\boldsymbol{\lambda}_{\text{next}} = \arg\max_{\boldsymbol{\lambda} \in \Lambda} \alpha(\boldsymbol{\lambda})$.

## 3.1 INR Configuration as a Global Optimization Problem

The central novelty of our work is to formalize the entire INR design process as a single, unified optimization problem. The performance of an INR is critically determined by the interplay between activation functions and weight initialization strategies on a layer-by-layer basis. Previous automated methods such as MIRE approach this by constructing networks greedily, selecting the best activation for each layer sequentially. This layer-wise greedy approach cannot guarantee global optimality, as the optimal choice for one layer is deeply conditioned on choices made for all other layers (see Theorem G.1 for details).

We instead define a global configuration vector $\boldsymbol{\Lambda}$ that simultaneously parameterizes choices for all $L$ layers of the network. For each layer $l \in \{1, \ldots, L\}$, we define a configuration tuple $\boldsymbol{\lambda}_l = (\sigma_l, \mathcal{I}_l, \mathbf{p}_l)$, where $\sigma_l \in \{\text{SIREN}, \text{WIRE}, \text{GAUSS}, \text{FINER++}, \text{FR}\}$ is a categorical variable for the activation function and $\mathcal{I}_l \in \{0, 1\}$ is a binary variable indicating the use of a SIREN-style initialization. The vector of continuous hyperparameters $\mathbf{p}_l \in \mathbb{R}^{d_p}$ amalgamates several crucial per-layer parameters: activation-specific values (e.g., frequency $\omega_0$ or scale $s_0$, conditional on the choice of $\sigma_l$), the initial range for the layer's weights, and a per-layer learning rate. The complete network configuration is the concatenation of these layer-wise tuples: $\boldsymbol{\Lambda}_{\text{network}} = (\boldsymbol{\lambda}_1, \boldsymbol{\lambda}_2, \ldots, \boldsymbol{\lambda}_L) \in \mathcal{L}$, where $\mathcal{L}$ denotes the high-dimensional, mixed-type configuration space. Our objective is to find the optimal configuration $\boldsymbol{\Lambda}^* = \arg\max_{\boldsymbol{\Lambda} \in \mathcal{L}} f(\boldsymbol{\Lambda})$, where $f(\boldsymbol{\Lambda})$ is the performance of the INR (e.g., validation PSNR) after being fully trained with the specified configuration. This evaluation constitutes the expensive black-box function we aim to optimize.

## 3.2 Surrogate Modeling of INR Configuration

**A Product Kernel for Mixed-Variable Spaces.** Our configuration vector $\boldsymbol{\Lambda}$ lives in a product space $\mathcal{X} = \mathcal{X}_{\text{cont}} \times \mathcal{X}_{\text{cat}}$, comprising continuous and categorical variables (Sheikh and Marcus, 2022; Lukovic et al., 2020). To model the correlation structure over this space, we design a product kernel that separates contributions from each variable type: $k(\boldsymbol{\Lambda}, \boldsymbol{\Lambda}') = k_{\text{cont}}(\boldsymbol{\Lambda}_c, \boldsymbol{\Lambda}_c') \times k_{\text{cat}}(\boldsymbol{\Lambda}_{\text{cat}}, \boldsymbol{\Lambda}_{\text{cat}}')$. For the continuous components $\boldsymbol{\Lambda}_c$, we use the Matérn kernel (Rasmussen and Williams, 2006; Daxberger et al., 2020), which generalizes the popular Squared Exponential (RBF) kernel and provides control over the smoothness of the surrogate function via parameter $\nu$: $k_{\text{cont}}(\boldsymbol{\Lambda}_c, \boldsymbol{\Lambda}_c') = \frac{2^{1-\nu}}{\Gamma(\nu)}(\sqrt{2\nu}\frac{||\boldsymbol{\Lambda}_c - \boldsymbol{\Lambda}_c'||_2}{\ell})^\nu K_\nu(\sqrt{2\nu}\frac{||\boldsymbol{\Lambda}_c - \boldsymbol{\Lambda}_c'||_2}{\ell})$, where $\ell$ is the length-scale and $K_\nu$ is the modified Bessel function. This flexibility is crucial for complex performance landscapes where the RBF kernel's assumption of infinite smoothness is often incorrect. For the categorical components $\boldsymbol{\Lambda}_{\text{cat}}$, we first transform them into a continuous space using one-hot encoding, where a categorical variable with $M$ levels is mapped to an $M$-dimensional binary vector. We then define $k_{\text{cat}}$ as a Squared Exponential kernel with Automatic Relevance Determination (ARD): $k_{\text{cat}}(\boldsymbol{\Lambda}_{\text{cat}}, \boldsymbol{\Lambda}_{\text{cat}}') = \exp(-\sum_{j=1}^{M} \frac{(\boldsymbol{\Lambda}_{\text{cat},j} - \boldsymbol{\Lambda}_{\text{cat},j}')^2}{2\ell_j^2})$, where each dimension has a unique length-scale $\ell_j$. The designed mechanism establishes the validity of our kernel ensures that as the number of evaluations grows, the posterior variance of the GP will concentrate around the true function $f(\boldsymbol{\Lambda})$ (see Theorem G.3 for details).

### 3.2.1 Empirical Expected Improvement via Matheron's Rule

The search for the next point to evaluate is guided by an acquisition function $\alpha : \mathcal{X} \to \mathbb{R}$ that balances exploration of uncertain regions with exploitation of promising areas. We adopt a Monte Carlo-based Empirical Expected Improvement (EEI) to overcome limitations of the analytic Expected Improvement (EI) function. While analytic EI admits a closed form for Gaussian posteriors in sequential settings, it becomes intractable for batch queries and exhibits sensitivity to model misspecification.

**Expected Improvement** Let $f : \mathcal{X} \to \mathbb{R}$ denote our objective function with GP prior $f \sim \mathcal{GP}(m_0, k_0)$. Given observations $\mathcal{D}_n = \{(\boldsymbol{\lambda}_i, y_i)\}_{i=1}^n$ where $y_i = f(\boldsymbol{\lambda}_i) + \epsilon_i$ with $\epsilon_i \sim \mathcal{N}(0, \sigma_n^2)$, and current best observation $f_{\text{best}} = \max_{i \in [n]} y_i$, the improvement function is defined as: $I(\boldsymbol{\lambda}) = \max\{0, f(\boldsymbol{\lambda}) - f_{\text{best}}\} = [f(\boldsymbol{\lambda}) - f_{\text{best}}]_+$. The Expected Improvement (Jones et al., 1998) is the expectation of this improvement under the posterior measure:

$$\text{EI}(\boldsymbol{\lambda}) = \mathbb{E}_{f(\boldsymbol{\lambda}) \sim p(\cdot|\mathcal{D}_n)}[I(\boldsymbol{\lambda})] = \int_{\mathbb{R}} [t - f_{\text{best}}]_+ \, p(f(\boldsymbol{\lambda}) = t|\mathcal{D}_n) \, dt$$

Under the GP posterior $f(\boldsymbol{\lambda})|\mathcal{D}_n \sim \mathcal{N}(\mu_p(\boldsymbol{\lambda}), \sigma_n^2(\boldsymbol{\lambda}))$, this admits the analytic form: $\text{EI}(\boldsymbol{\lambda}) = \sigma_n(\boldsymbol{\lambda})[\phi(Z)\Phi(Z) + Z]$ where $Z = \frac{\mu_n(\boldsymbol{\lambda}) - f_{\text{best}}}{\sigma_n(\boldsymbol{\lambda})}$, and $\phi$, $\Phi$ denote the standard normal PDF and CDF respectively.

**Monte Carlo Approximation.** For batch optimization and robustness to model misspecification, we employ a Monte Carlo estimator. Let $\{f^{(s)}\}_{s=1}^S$ be i.i.d. samples from the posterior process. The Empirical Expected Improvement (Wilson et al., 2018) is:

$$\widehat{\text{EI}}(\boldsymbol{\lambda}) = \frac{1}{S} \sum_{s=1}^S [f^{(s)}(\boldsymbol{\lambda}) - f_{\text{best}}]_+$$

By the Strong Law of Large Numbers, $\widehat{\text{EI}}(\boldsymbol{\lambda}) \xrightarrow{a.s.} \text{EI}(\boldsymbol{\lambda})$ as $S \to \infty$. The convergence rate follows $\mathbb{E}[|\widehat{\text{EI}}(\boldsymbol{\lambda}) - \text{EI}(\boldsymbol{\lambda})|^2] = \mathcal{O}(S^{-1})$ by the Central Limit Theorem.

**Efficient Posterior Sampling via Matheron's Rule.** Direct posterior sampling requires computing the Cholesky decomposition of $\mathbf{K}_n + \sigma_n^2\mathbf{I} \in \mathbb{R}^{n \times n}$, incurring $\mathcal{O}(n^3)$ cost per sample. For $S$ samples, this yields prohibitive $\mathcal{O}(Sn^3)$ complexity. Alternative approaches such as random Fourier features or sparse GPs sacrifice posterior accuracy, which is critical for our high-dimensional optimization problem. We leverage Matheron's rule (Rasmussen and Williams, 2006; Daulton et al., 2022) (also known as the conditional simulation formula) for exact posterior sampling with dramatically reduced computational cost. This approach offers critical advantages over alternative methods. First, it provides exceptional computational efficiency by computing the expensive matrix inversion $[\mathbf{K} + \sigma_n^2\mathbf{I}]^{-1}$ only once, reducing complexity from $\mathcal{O}(Sn^3)$ to $\mathcal{O}(n^3 + Sn^2)$ for $S$ samples (see Theorem G.4 for details). Second, unlike approximation methods such as inducing points or random features, Matheron's rule produces exact samples from the true posterior distribution, preserving the GP's uncertainty quantification that is crucial for balancing exploration and exploitation in our optimization problem. Third, once the weight vector $\mathbf{w}$ is computed, posterior evaluations at different points can be parallelized across samples and query locations, enabling efficient GPU utilization and further accelerating the optimization process. Algorithm 1 in Section A outlines the complete workflow for discovering optimal INR configurations with our Bayesian optimization framework.

**Theorem (Matheron's Rule):** Let $f \sim \mathcal{GP}(m_0, k_0)$ be a GP prior and $\mathcal{D}_n = \{(\mathbf{X}, \mathbf{y})\}$ be observations. A sample from the posterior process can be expressed as: $f_{\text{post}}(\cdot) \overset{d}{=} f_{\text{prior}}(\cdot) + \mathbf{k}(\cdot, \mathbf{X})[\mathbf{K} + \sigma_n^2\mathbf{I}]^{-1}(\mathbf{y} - f_{\text{prior}}(\mathbf{X}))$, where $f_{\text{prior}} \sim \mathcal{GP}(m_0, k_0)$, $\mathbf{K}_{ij} = k_0(\mathbf{x}_i, \mathbf{x}_j)$, and $\overset{d}{=}$ denotes equality in distribution. This decomposition enables the following efficient sampling procedure: first, draw one sample path $f_{\text{prior}} \sim \mathcal{GP}(m_0, k_0)$ using random Fourier features or inducing points; second, compute the weight vector $\mathbf{w} = [\mathbf{K} + \sigma_n^2\mathbf{I}]^{-1}(\mathbf{y} - f_{\text{prior}}(\mathbf{X}))$ once; third, for any query point $\boldsymbol{\lambda}$, compute $f_{\text{post}}(\boldsymbol{\lambda}) = f_{\text{prior}}(\boldsymbol{\lambda}) + \mathbf{k}(\boldsymbol{\lambda}, \mathbf{X})\mathbf{w}$ The computational complexity is $\mathcal{O}(n^3)$ for the initial matrix inversion plus $\mathcal{O}(n)$ per query point evaluation, amortizing the cost across $S$ samples. This methodology provides a principled, globally-aware strategy for exploring the mixed-variable configuration space $\mathcal{X}$, capturing complex interdependencies between layers, activation functions, and initialization schemes to discover high-performing architectures in a fully automated fashion.

# 4 Related Work

Our work builds upon three core areas of research: the development of implicit neural representations, the design of specialized activation functions to overcome spectral bias, and the application of automated machine learning to architectural design. **Implicit Neural Representations.** The paradigm of representing signals as continuous functions parameterized by coordinate-based MLPs has fundamentally reshaped fields like 3D vision and computer graphics Li et al. (2021); Xie et al. (2022). Foundational works such as DeepSDF Park et al. (2019) and Occupancy Networks Mescheder et al. (2019) demonstrated the efficacy of INRs for high-fidelity 3D shape modeling. This was famously extended to novel view synthesis with Neural Radiance Fields (NeRF) Mildenhall et al. (2020), cementing INRs as a powerful, resolution-agnostic alternative to traditional discrete representations. **Activation Functions and Spectral Bias.** A primary challenge in training INRs is the inherent spectral bias of standard MLPs, which struggle to learn high-frequency functions Rahaman et al. (2019). Early solutions relied on positional encoding with Fourier features to inject high-frequency information at the input layer Tancik et al. (2020). A significant breakthrough came with Sinusoidal

Representation Networks (SIRENs) Sitzmann et al. (2020a), which showed that using periodic activation functions throughout the network could natively represent fine details. The success of SIREN spurred an explosion of research into alternative activation functions, each with a unique inductive bias, including wavelet-based (WIRE) Saragadam et al. (2023), Gaussian Ramasinghe and Lucey (2022), and variable-periodic (FINER) Liu et al. (2024) activations. While this has created a rich toolkit, it has also transformed INR design into a complex configuration problem where performance is highly sensitive to the choice of activation and its initialization. **Automated Configuration for INRs.** Our work addresses this challenge by drawing from the principles of Automated Machine Learning (AutoML) and Neural Architecture Search (NAS) Elsken et al. (2019); Feurer and Hutter (2019). We employ Bayesian optimization, a sample-efficient global optimization strategy well-suited for expensive black-box functions like training a neural network Snoek et al. (2012). While most INR research relies on manual tuning, the most relevant automated approach is MIRE Jayasundara et al. (2025), which uses a greedy, layer-wise dictionary learning method to select activations. However, its sequential nature cannot guarantee global optimality. Our framework, `OptiINR`, distinguishes itself by performing a global, joint optimization over all layers simultaneously. This approach is also distinct from meta-learning frameworks like MetaSDF Sitzmann et al. (2020b); Tancik et al. (2021), which learn priors for fast adaptation to new signals, whereas our goal is to find the single best-performing architecture for a specific, individual signal.

## 5 Experiments

To rigorously validate the OptiINR framework, we designed a comprehensive suite of experiments aimed at answering three central research questions. First, can a principled, global optimization framework discover configurations that consistently and significantly outperform state-of-the-art, manually-tuned baselines across diverse signal modalities? Second, does the framework's efficacy scale from low-dimensional signals to more complex, high-dimensional representations? Third, and most critically, can the architectures discovered through automated search reveal novel, generalizable design principles that challenge or refine conventional heuristics in INR design? Through meticulous quantitative and qualitative analysis across multiple canonical tasks (Sitzmann et al., 2020a; Saragadam et al., 2023; Liu et al., 2024), we demonstrate that OptiINR not only automates and elevates the configuration process but also serves as a powerful tool for advancing the fundamental understanding of what constitutes an optimal implicit neural representation.

**Experimental Protocol** All experiments were conducted using PyTorch (Paszke et al., 2019), with the Bayesian optimization component implemented via the BoTorch library (Balandat et al., 2020). To isolate the impact of network configuration, we employed a consistent base architecture across all evaluated models: a four-layer MLP with 256 hidden units per layer. Each configuration was trained for 10,000 epochs using the AdamW optimizer (Loshchilov and Hutter, 2017) with learning rate $1 \times 10^{-4}$, without learning rate scheduling. All evaluations were performed on NVIDIA B200 GPUs. This standardized setup ensures that performance differences are attributable solely to the configuration—the layer-wise combination of activations and initializations—which is the primary variable under investigation.

**OptiINR Configuration Space:** The core of our method is the structured, mixed-variable search space, which OptiINR navigates to find optimal configurations. This space encompasses critical design choices for a 4-layer network architecture. A binary variable determines whether to use standard Fourier feature positional encoding (Tancik et al., 2020), with a corresponding continuous parameter if PE is used, controlling the scale of input coordinate mapping. For each of the four hidden layers, a categorical variable selects the activation function from task-specific sets: {SIREN, FINER } for audio representation, {SIREN, FINER, FINER++, WIRE } for image representation, and {FINER, Gauss, FINER++, WIRE } for 3D shape representation, enabling discovery of heterogeneous architectures tailored to each signal modality. Each selected activation function has an associated continuous hyperparameter (e.g., $\omega_0$) that is jointly optimized, allowing fine-tuning of the activation's spectral properties on a per-layer basis. Additionally, to account for varying optimization dynamics across network depth, each of the four layers has its own independent learning rate $\alpha_l$ optimized as a continuous variable.

**Baseline Methods:** To ensure rigorous and fair comparison, OptiINR was evaluated against a comprehensive set of state-of-the-art INRs using their officially published or standard configurations, measuring OptiINR against methods operating under their ideal, author-optimized conditions. The selected baselines represent diverse inductive biases: SIREN (Sitzmann et al., 2020a), the foundational model employing periodic sinusoidal activations; FINER (Liu et al., 2024), a recent advance using

variable-periodic activations for flexible spectral control; Gauss (Ramasinghe and Lucey, 2022), a representative non-periodic activation based on locality; Wavelet (WIRE) (Saragadam et al., 2023), a robust model based on complex Gabor wavelets known for excellent space-frequency localization; FR (Zheng et al., 2024), a recent method based on Fourier reparameterized training; and IGA (Zheng et al., 2024), an improved SIREN variant incorporating inductive gradient adjustment. Our OptiINR optimization process began with 30 initial configurations generated via space-filling Latin Hypercube sampling to ensure broad initial coverage, followed by 100 iterations of Bayesian optimization to refine the search and discover optimal configurations through automated exploration-exploitation balancing.

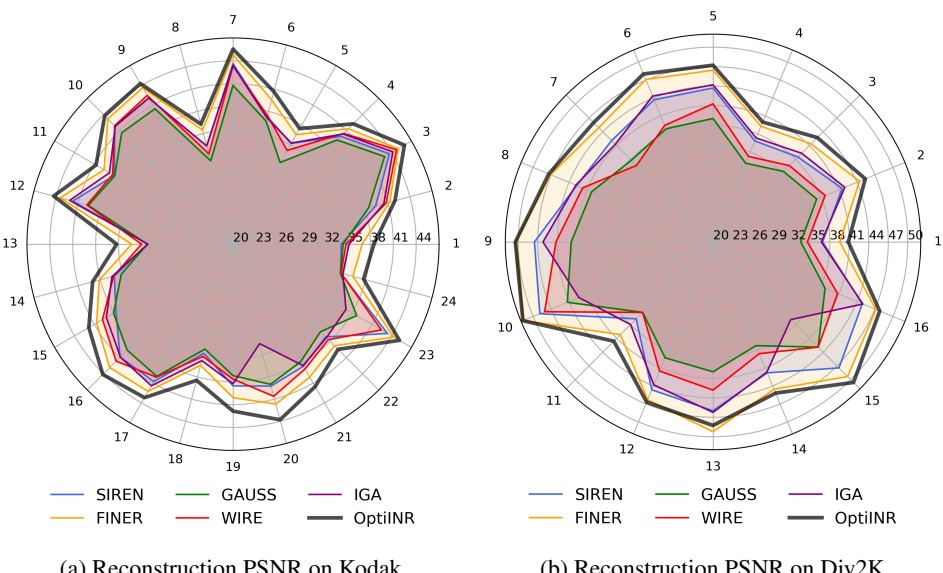

(a) Reconstruction PSNR on Kodak      (b) Reconstruction PSNR on Div2K

Figure 1: Detailed per-image PSNR comparisons across all methods on Kodak and Div2K

### 5.1 Image Representation

Image representation serves as the canonical benchmark for INR capabilities, requiring networks to learn continuous mappings $f : \mathbb{R}^2 \to \mathbb{R}^3$ from pixel coordinates to RGB values. This task challenges INRs to capture both smooth gradients and high-frequency details present in natural images, making it an ideal testbed for configuration optimization. Networks are provided with normalized coordinates without positional embedding and trained to predict corresponding RGB values over 10,000 epochs.

**Datasets and Evaluation Protocol.** We evaluate on two complementary benchmarks: the Kodak dataset (Franzen, 1999) containing 24 diverse natural images at $768 \times 512$ resolution encompassing portraits, landscapes, architecture, and detailed textures; and the DIV2K dataset (Agustsson and Timofte, 2017), where we use 16 cropped $512 \times 512$ patches selected for varied texture complexities and frequency characteristics, providing a challenging high-resolution testbed.

Table 1: Average PSNR (dB) $\pm$ std on image representation tasks. OptiINR consistently outperforms all baselines.

| Method | Kodak | DIV2K |
|---|---|---|
| SIREN | $38.47 \pm 3.47$ | $42.75 \pm 3.91$ |
| Gauss | $37.36 \pm 3.11$ | $38.48 \pm 3.13$ |
| WIRE | $38.69 \pm 3.50$ | $39.85 \pm 3.81$ |
| FR | $35.90 \pm 2.42$ | $38.87 \pm 2.27$ |
| FINER | $40.24 \pm 3.23$ | $45.56 \pm 3.84$ |
| GF | $38.47 \pm 4.50$ | $40.57 \pm 5.54$ |
| IGA | $38.27 \pm 3.43$ | $41.77 \pm 3.24$ |
| **OptiINR (ours)** | $\mathbf{41.38 \pm 3.05}$ | $\mathbf{46.24 \pm 3.49}$ |

**Quantitative Results.** Table 1 summarizes the average PSNR and standard deviation across both datasets, demonstrating OptiINR's substantial performance gains. On Kodak, OptiINR achieves 41.38 dB average PSNR, surpassing the strongest baseline FINER by 1.14 dB and showing remarkable improvements over SIREN (2.91 dB), Gaussian activations (4.02 dB), and Fourier Reparameterization (5.48 dB). Figure 1 presents the detailed per-image PSNR comparisons across all methods, revealing

that improvements are consistent across all 24 Kodak images without exception, with per-image gains ranging from 0.91 to 4.14 dB over the best baseline for each image.

On DIV2K's high-resolution patches, OptiINR demonstrates even more pronounced advantages, achieving PSNR values from 39.99 to an exceptional 51.70 dB as shown in Table 1. The average 46.24 dB represents approximately 3–4 dB improvement over the best baselines, with particularly dramatic gains on images containing repetitive patterns or fine details where traditional INR activations fail to capture the full frequency spectrum.

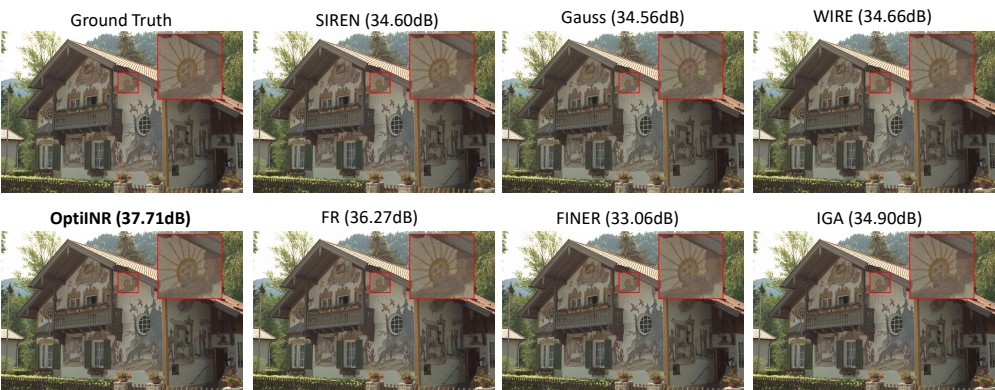

Figure 2: Kodak 24 with the region of interest (red box) and an upper-right enlargement rendered with nearest-neighbor to preserve pixel details. All methods use the same ROI for fair visual comparison.

**Configuration Adaptation Analysis.** OptiINR's discovered configurations reveal sophisticated adaptation to image characteristics. This is visualized qualitatively for two representative images from the Kodak dataset in Figure 2, which shows the final reconstructions. To further highlight the performance differences, Figure 3 and Figure 4 display the corresponding error fields for all evaluated methods. For smooth, low-frequency content, OptiINR selects Gaussian or FINER++ activations in early layers for smooth interpolation, followed by periodic activations (SIREN, sinusoidal) in deeper layers to capture residual high-frequency components. For texture-rich images with prominent edges, OptiINR favors wavelet-based activations (WIRE, Gabor) throughout the network, leveraging their optimal space-frequency localization. This automatic adaptation eliminates manual parameter tuning where single misconfigurations can degrade performance by several dB. Notably, OptiINR discovers novel activation combinations unexplored in prior work, such as using band-limited functions in intermediate layers to bridge spatially-localized early features and globally periodic final layers. This leads to the superior reconstructions shown in Figure 2, where the reduction in reconstruction error is made evident by the significantly attenuated error fields in Figures 3 and 4.

## 5.2 Audio Reconstruction

Audio reconstruction presents unique challenges for INRs, requiring precise capture of temporal dynamics as demonstrated in HyperSound (Szatkowski et al., 2023), harmonic relationships, and frequency content spanning multiple octaves. The task is formulated as learning a mapping $f : \mathbb{R} \to \mathbb{R}$ from time coordinates to signal amplitude, where the network must represent complex waveforms with extremely high-frequency details and intricate harmonic structures.

**Datasets and Evaluation Protocol.** We evaluate on three standard audio signals from the SIREN (Sitzmann et al., 2020a) benchmark: Bach (complex polyphonic composition with intricate harmonic structures), Counting (speech with distinct phonetic transitions), and Two Speakers (overlapping voices requiring separation of distinct characteristics). Following established protocols, the output layer was initialized with $\mathcal{U}(-10^{-4}, 10^{-4})$ distribution

Table 2: PSNR (dB) comparison on audio reconstruction. OptiINR achieves breakthrough performance.

| Method | Bach | Count | Two Spk |
|--------|------|-------|---------|
| SIREN | 52.59 | 34.39 | 41.59 |
| Gauss | 16.49 | 21.32 | 17.21 |
| WIRE | 17.54 | 21.54 | 24.16 |
| FR | 54.94 | 36.93 | 56.36 |
| FINER | 36.67 | 39.35 | 42.27 |
| IGA | 52.35 | 34.41 | 42.39 |
| **OptiINR** | **60.84** | **49.60** | **68.39** |

and zero biases for stable training, input coordinates were mapped to $[-100, 100]$, and models were trained for 10,000 iterations.

**Quantitative Results.** Table 2 demonstrates OptiINR's exceptional performance gains across all audio signals. On the Bach composition, OptiINR achieves 60.84 dB PSNR, surpassing the best baseline (FR) by 5.90 dB and SIREN by 8.25 dB. The Counting sequence sees OptiINR reaching 49.60 dB versus FINER's 39.35 dB—a remarkable 10.25 dB improvement. Most dramatically, on the Two Speakers signal, OptiINR achieves 68.39 dB compared to FR's 56.36 dB, representing a 12.03 dB gain. These substantial numerical improvements translate to orders-of-magnitude differences in reconstruction error, with OptiINR achieving a near-machine-precision loss ($\approx 10^{-6}$) while baselines struggle with losses 3–4 orders of magnitude higher. This exceptional accuracy is visualized in Figure 5 and 6, which present a detailed comparison of the reconstructed waveforms and their corresponding spectral analyses. The predicted audio signal from OptiINR is visually indistinguishable from the ground truth waveform, perfectly capturing the amplitude and temporal dynamics. In contrast, baseline methods exhibit significant distortions, failing to replicate the signal's structure with high fidelity. The spectrum analysis further confirms this superiority; the signed spectral residual plot for OptiINR is almost entirely neutral, indicating a near-perfect match to the ground truth spectrum across all frequencies. Baselines, however, show large regions of spectral error, demonstrating their inability to accurately reconstruct the full frequency content. This exceptional accuracy allows OptiINR to preserve subtle audio characteristics, including room acoustics, instrumental timbres, and voice inflections that are completely lost in baseline reconstructions.

### 5.3 3D Shape Representation: Occupancy Reconstruction

Three-dimensional shape representation through occupancy fields tests INRs' ability to model complex geometric structures and maintain topological consistency across multiple spatial scales. This task involves learning a function $f : \mathbb{R}^3 \to \{0, 1\}$ following the occupancy network formulation (Mescheder et al., 2019) that maps voxel coordinates to binary occupancy values, where 1 indicates object presence and 0 denotes empty space, effectively acting as a 3D point classifier.

**Dataset and Experimental Setup.** We evaluate on high-resolution models from the Stanford 3D Scanning Repository (Levoy et al., 2000): the Dragon and Thai Statue, chosen for their intricate geometric details and varied surface characteristics. Both models were voxelized at $512^3$ resolution, providing a challenging testbed for precise boundary representation. Performance is measured using Intersection over Union (IoU), which captures occupancy quality while ignoring the large number of trivial true negatives.

Table 3: IoU comparison on 3D occupancy reconstruction at $512^3$ resolution.

| Method | Dragon | Thai Statue |
|--------|--------|-------------|
| SIREN | 0.9881 | 0.9778 |
| Gauss | 0.9934 | 0.9871 |
| WIRE | 0.9924 | 0.9861 |
| FR | 0.9919 | 0.9650 |
| FINER | 0.9897 | 0.9804 |
| IGA | 0.9919 | 0.9834 |
| **OptiINR** | **0.9936** | **0.9884** |

**Quantitative results.** Table 3 shows OptiINR's consistent gains in geometric accuracy. On Dragon, OptiINR attains 0.9936 IoU vs. 0.9934 for the best baseline (Gaussian activations); while a 0.0002 absolute gain appears small, on a $512^3$ grid it corresponds to $\approx 2.7 \times 10^4$ additional correct voxel decisions, concentrated in high-curvature regions (scales, wing membranes, facial details). On Thai Statue, OptiINR reaches 0.9884 IoU vs. 0.9871, with improvements primarily on carved motifs and thin protrusions requiring precise localization. Reconstruction visualizations are provided in Fig. 7 and Fig. 8.

## 6 Conclusion

Configuring implicit neural representations (INRs) is increasingly challenging, so we recast it as a global optimization problem rather than relying on manual tuning and ad-hoc heuristics. OptiINR uses Bayesian optimization to jointly select activation functions and initialization schemes, yielding a unified, sample-efficient, architecture-agnostic procedure. Across core applications—2D image representation, 3D shape modeling, and novel-view synthesis—configurations discovered by OptiINR consistently outperform state-of-the-art manual baselines and prior automated methods. Analysis shows the optimal design is strongly task-dependent, revealing the limits of one-size-fits-all rules and motivating principled automated search. By providing an extensible foundation for INR design, OptiINR improves performance and reliability, scales with evaluation budgets, and helps close the capacity–convergence gap that has constrained practical effectiveness.

## Ethics Statement

We have read, understand, and agree to abide by the ICLR Code of Ethics for all aspects of this work (submission, authorship, and discussion). Our study proposes methodological advances in Bayesian optimization for implicit neural representations and is evaluated exclusively on publicly available, non-sensitive datasets (Kodak/DIV2K images, SIREN audio benchmarks, and Stanford 3D models). No human-subjects data, personally identifiable information, or user-generated private content are used; accordingly, IRB approval was not required. We comply with all dataset licenses and do not redistribute copyrighted data; instead, we will provide scripts to download sources from their official repositories together with clear preprocessing documentation. We are not aware of conflicts of interest or sponsorship that could bias the work. Potential risks include dual-use of improved reconstruction fidelity (e.g., circumventing image/audio protections); to mitigate this, we will release code for research purposes under a standard academic license, include a responsible-use notice, and refrain from providing artifacts designed to remove watermarks or bypass access controls. Given the non-demographic, non-sensitive nature of the benchmarks, fairness and discrimination risks are minimal; nevertheless, we caution against deploying our methods in downstream applications where such harms could arise without appropriate auditing. We report hardware, training budgets, and random seeds to reduce unnecessary reruns and limit environmental impact. All results are documented to support research integrity (complete references, clear assumptions, and reproducible procedures).

## Reproducibility Statement

We have organized all information needed to reproduce our results across the main paper, appendix, and anonymous supplementary materials. The overall methodology and algorithmic workflow (including the GP surrogate over mixed variables and the empirical EI via Matheron's rule) are described in Section 3, with complete algorithmic pseudocode in Appendix A (Algorithms 1–2). The mixed-variable kernel construction and acquisition strategy are detailed in Sections 3.2 and 3.2.1, respectively. Our Experimental Protocol (Section 5) specifies the search space (activation sets and hyperparameter ranges), training configurations (model architecture, optimizer, schedules, budgets), tooling and hardware, and the canonical seeds/budgets used in all experiments; the section titled OptiINR Configuration Space and Baseline Methods further document baseline setups and the Bayesian optimization procedure (initialization strategy and iteration budgets). Task-specific datasets, preprocessing steps, metrics, and evaluation procedures are provided in Sections 5.1–5.3 (e.g., image representation on Kodak/DIV2K with PSNR, audio reconstruction with STFT-based evaluation, and 3D occupancy with IoU), and visualization/metric pipelines (e.g., residual heatmaps and signal analyses) are summarized in Appendix B. Baseline configurations for SIREN, FINER, GAUSS, WIRE, FR, and IGA follow published defaults as cited in the main text.

## The Use of Large Language Models (LLMs)

We used an LLM-based assistant solely for copy-editing and phrasing improvements. The model did not generate research ideas, design experiments, analyze data, or contribute substantive content. All technical contributions and conclusions are the authors' own; all edits were reviewed and verified by the authors. No confidential data beyond the manuscript text were provided, and this usage complies with the ICLR Code of Ethics.

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

# A  OptiINR Algorithm

The full workflow for discovering optimal INR configurations is outlined in Algorithm 1. The process iteratively refines its model of the performance landscape and makes increasingly informed decisions. The procedure for maximizing the acquisition function is detailed in Algorithm 2.

---

**Algorithm 1** OptiINR: Bayesian Optimization for INR Configuration

---

1: **Input:** Objective function $f(\cdot)$, search space $\mathcal{L}$, initial samples $N_{\text{init}}$, total iterations $T$.
2: **Initialize:** GP with mixed-variable product kernel $k(\cdot, \cdot')$.
3: **Initialization Phase:**
4:     Sample initial configurations $\{\mathbf{\Lambda}_i\}_{i=1}^{N_{\text{init}}}$ from $\mathcal{L}$ using a space-filling design.
5:     Evaluate the objective function for each initial configuration: $\mathcal{D}_{\text{init}} = \{(\mathbf{\Lambda}_i, f(\mathbf{\Lambda}_i))\}_{i=1}^{N_{\text{init}}}$.
6: **Optimization Loop:**
7: **for** $t = N_{\text{init}}$ to $T - 1$ **do**
8:     Fit GP surrogate model to the current dataset $\mathcal{D}_t$.
9:     Find next configuration by maximizing Empirical Expected Improvement (see Algorithm 2):
        $\mathbf{\Lambda}_{t+1} = \arg\max_{\mathbf{\Lambda} \in \mathcal{L}} \widehat{\text{EI}}(\mathbf{\Lambda}|\mathcal{D}_t)$.
10:     Evaluate objective: $y_{t+1} = f(\mathbf{\Lambda}_{t+1})$.
11:     Update dataset: $\mathcal{D}_{t+1} = \mathcal{D}_t \cup \{(\mathbf{\Lambda}_{t+1}, y_{t+1})\}$.
12: **end for**
13: **Return:** $\mathbf{\Lambda}^* = \arg\max_{(\mathbf{\Lambda}, y) \in \mathcal{D}_T} y$.

---

**Algorithm 2** Empirical Expected Improvement (EEI) Computation

---

1: **Input:** Candidate configuration $\mathbf{\Lambda}$, GP posterior from data $\mathcal{D}_t = \{(\mathbf{X}, \mathbf{y})\}$, best value $y_{best}$, number of samples $S$.
2: **Define:** GP prior $f_{prior} \sim \mathcal{GP}(0, k)$.
3: **Pre-computation:**
4:     Compute matrix inverse $\mathbf{W} = [k(\mathbf{X}, \mathbf{X}) + \sigma_n^2 \mathbf{I}]^{-1}$.
5: **Monte Carlo Estimation:**
6:     Initialize total improvement $I_{total} = 0$.
7: **for** $s = 1$ to $S$ **do**
8:     Draw a sample function from the GP prior: $f_{prior}^{(s)} \sim \mathcal{GP}(0, k)$.
9:     Evaluate prior sample at observed data points: $\mathbf{y}_{prior}^{(s)} = f_{prior}^{(s)}(\mathbf{X})$.
10:     Evaluate prior sample at candidate point: $y_{cand\_prior}^{(s)} = f_{prior}^{(s)}(\mathbf{\Lambda})$.
11:     Generate posterior sample using Matheron's rule:
        $y_{post}^{(s)} = y_{cand\_prior}^{(s)} + k(\mathbf{\Lambda}, \mathbf{X})\mathbf{W}(\mathbf{y} - \mathbf{y}_{prior}^{(s)})$.
12:     Calculate improvement for the sample: $I_s = \max(0, y_{post}^{(s)} - y_{best})$.
13:     Accumulate improvement: $I_{total} = I_{total} + I_s$.
14: **end for**
15: **Return:** Estimated EEI: $\widehat{\text{EI}}(\mathbf{\Lambda}) = I_{total}/S$.

---

# B  Error Fields of Image Representation

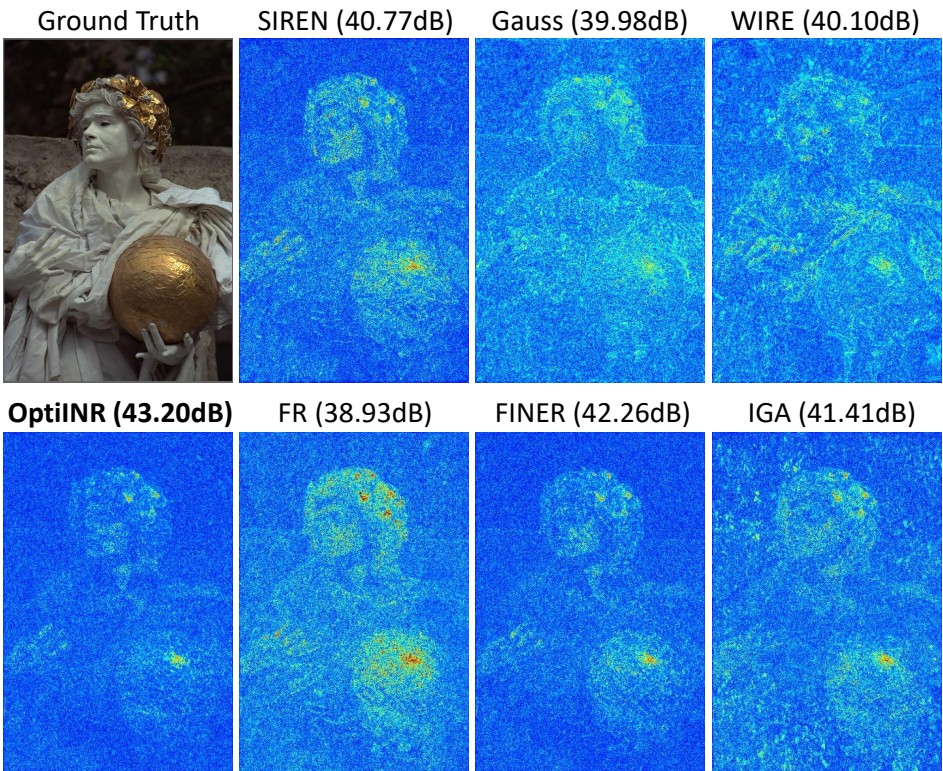

Figure 3: Residual heatmap visualization on Kodak 17 with respect to the reference image. For each baseline, we compute per-pixel absolute differences to the reference (averaged over RGB), normalize them to $[0, 1]$, and enhance visibility using gain (GAIN=16) and gamma ($\gamma$=0.6). The residuals are colorized using the jet colormap, where blue indicates low error and red indicates high error, and they are overlaid on the reconstructed image with an opacity of 0.85.

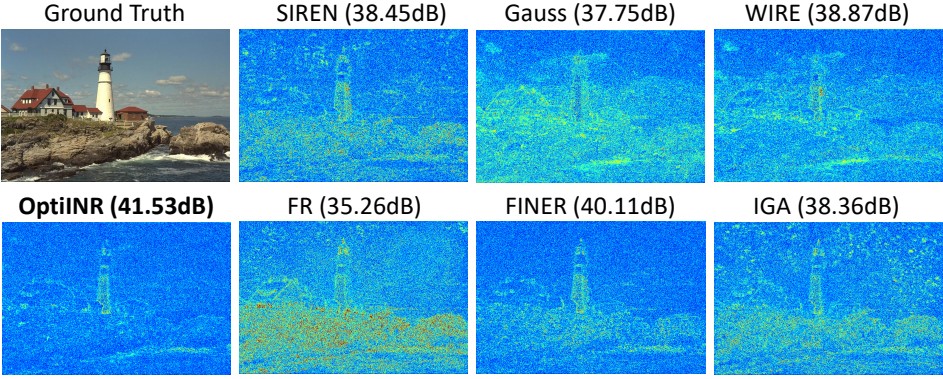

Figure 4: Residual heatmap visualization on Kodak 21 with respect to the reference image. For each baseline, we compute per-pixel absolute differences to the reference (averaged over RGB), normalize them to $[0, 1]$, and enhance visibility using gain (GAIN=16) and gamma ($\gamma$=0.6). The residuals are colorized using the jet colormap, where blue indicates low error and red indicates high error, and they are overlaid on the reconstructed image with an opacity of 0.85.

## C   Spectral Analysis of Audio Reconstruction

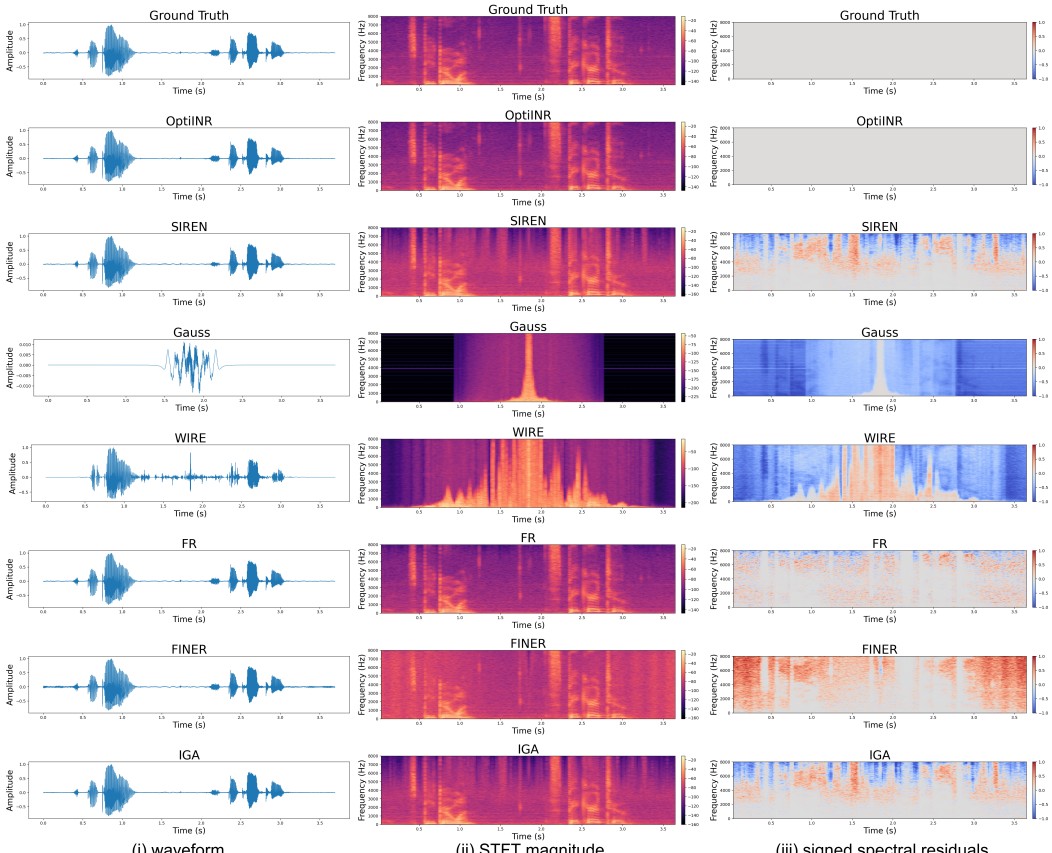

Figure 5: Columns show: (i) waveform, (ii) STFT magnitude (in dB), and (iii) signed spectral residuals. Rows (top to bottom) correspond to: Ground Truth, OptiINR, SIREN, Gauss, WIRE, FR, FINER, and IGA. The experiment is conducted on the `TwoSpeakers` dataset. The STFT was computed using a Hann window with a frame length of 1024 samples and a hop size of 256 samples, and results are visualized with a `magma` colormap. Residual maps are obtained by subtracting the reference STFT (in dB) from the test STFT (in dB), followed by 99.5% percentile clipping, a gain of 1.0, and gamma correction of 0.9. Residual heatmaps use a zero-centered diverging colormap, where blue indicates regions where the reference has stronger energy and red indicates regions where the test signal has stronger energy.

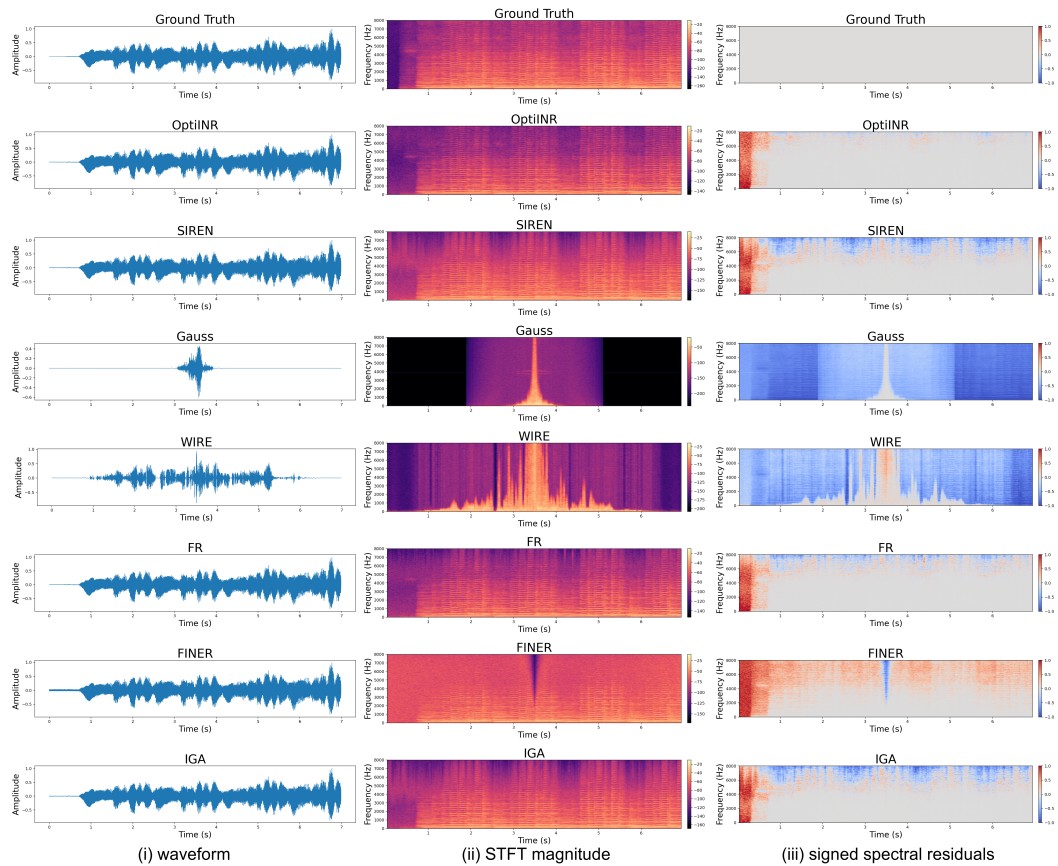

Figure 6: Same setting as Fig. 5, but on the `Bach` dataset.

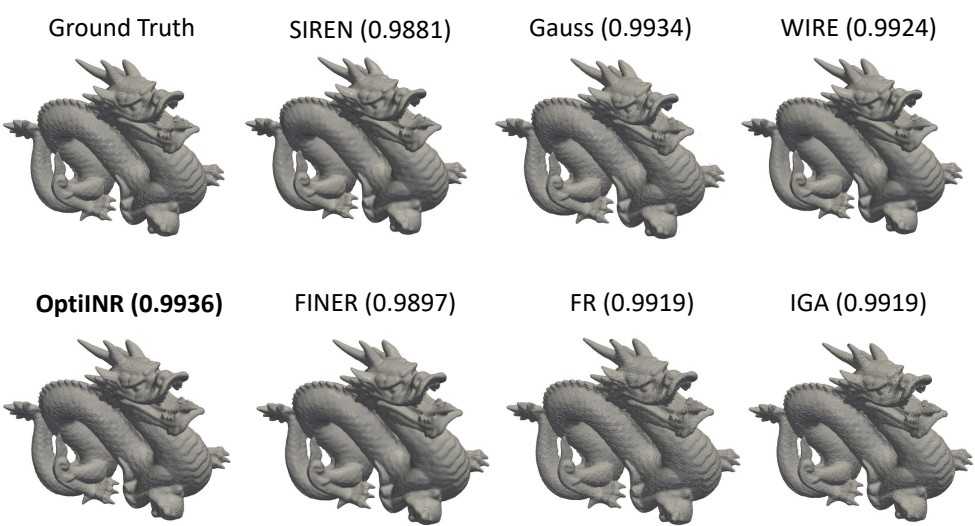

Figure 7: Visualization of 3D dataset Dragon

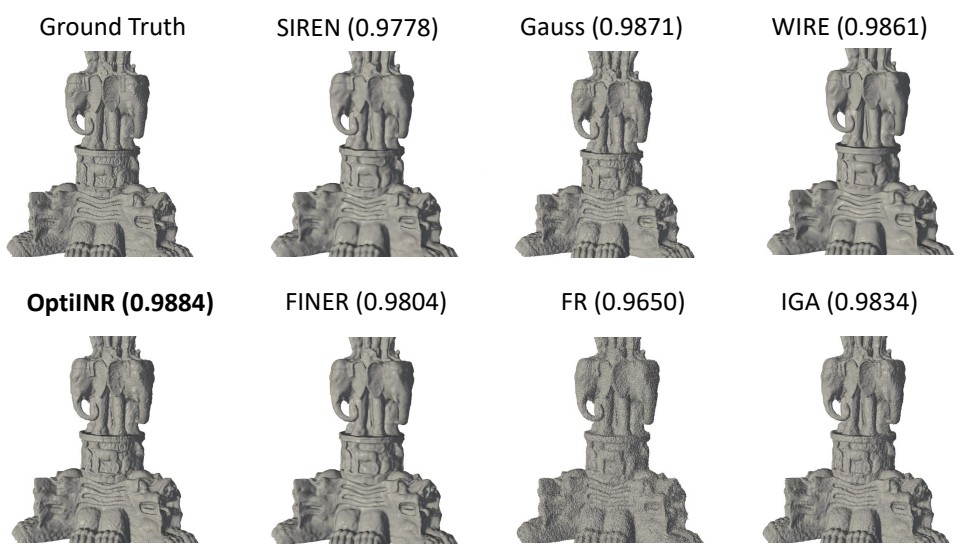

Figure 8: Visualization of 3D dataset Thai Statue

# D  NEURAL RADIANCE FIELDS

**Dataset and Experimental Setup.** We evaluate OptiINR on the Lego scene from the synthetic NeRF dataset using the vanilla implementation of Mildenhall et al. (2020). All methods use the original 8-layer, 256-width MLP. Following standard NeRF practice, we apply positional encoding (PE) with 10 frequencies for 3D coordinates (`multires=10`) and 4 frequencies for view directions (`multires_views=4`).

Table 4: PSNR comparison on NeRF reconstruction.

| Method | PSNR ↑ |
| --- | --- |
| ReLU+PE | 25.05 |
| SIREN | 23.87 |
| FINER | 24.25 |
| Gauss | 22.25 |
| WIRE | 24.63 |
| **OptiINR** | **25.63** |

To ensure a fair comparison across INR activations, we unify the optimizer settings by training all baselines with Adam at a fixed learning rate of $5 \times 10^{-5}$ for 30,000 iterations while keeping the sampling pipeline, ray-marching hyperparameters, and dataset splits unchanged (100 train views, 13 validation views, 25 test views). ReLU uses the standard NeRF positional encoding, while other INR activations follow the established practice in INR reconstruction benchmarks and are evaluated both without PE (their default setting) and under OptiINR's BO-driven configuration search. OptiINR performs BO over the configuration space consisting of PE usage, per-layer activation families and initialization parameters, and layerwise learning rates.

**Quantitative Results.** Table 4 reports PSNR on held-out test views. When trained with unified settings, ReLU+PE reaches 25.05 dB, substantially stronger than the classical INR baselines without PE (SIREN 23.87 dB, Gauss 22.25 dB, FINER 24.25 dB, and WIRE 24.63 dB). This aligns with prior observations that PE is essential for ReLU-based NeRFs but does not trivially benefit INR activations designed to encode high frequencies directly.

OptiINR achieves 25.63 dB, outperforming the strongest baseline (WIRE, 24.63 dB) by 1.00 dB and surpassing the improved ReLU+PE baseline by 0.58 dB. In NeRF benchmarks—where architectural or sampling changes typically yield only 0.3–0.5 dB improvements—this 0.6–1.0 dB margin obtained purely from configuration optimization is substantial. These results indicate that activation and initialization choices are a critical yet under-explored component of radiance-field modeling.

**Configuration Adaptation Analysis.** The configurations discovered by OptiINR show a consistent pattern across trials. Early layers tend to adopt smoother or band-limited activations to improve coarse geometry stability under volumetric rendering, whereas deeper layers select more oscillatory or directional activations (e.g., SIREN- or WIRE-like families) to capture fine view-dependent effects. BO also identifies a non-uniform layerwise learning-rate pattern, assigning larger rates to shallow layers for rapid global structure fitting and smaller rates to deeper layers for stable refinement of high-frequency appearance. These adaptations arise automatically without modifying the NeRF architecture or sampling process, illustrating OptiINR's ability to uncover non-trivial, task-dependent INR configurations.

# E  PDE Reconstruction: Shear Flow

**Dataset and Experimental Setup.** To evaluate OptiINR on physics-based signals, we conduct a PDE reconstruction experiment using the `shear_flow` dataset from The Well benchmark. Each simulation provides time-resolved tracer, pressure, and velocity fields. Following our preprocessing pipeline, we extract only the tracer field, preserve all temporal steps, and spatially downsample each trajectory to $64 \times 64$ using bilinear interpolation. This yields spatiotemporal tensors of shape $(51, 64, 64)$ for each trajectory.

Although no PDE supervision or physics constraints are used during training, the underlying tracer obeys the advection–diffusion equation

$$\partial_t s - D\Delta s = -u \cdot \nabla s,$$

where $u = (u_x, u_y)$ is the velocity field and $D$ is a diffusivity parameter determined by the Reynolds and Schmidt numbers. This PDE generates sharp advective fronts and filament structures, making the tracer field a strong benchmark for INR expressiveness.

We reconstruct the full spatiotemporal tracer field from a single trajectory (trajectory #30) using the INR formulation

$$f_\theta : (t, x, y) \in \mathbb{R}^3 \to \mathbb{R},$$

where normalized coordinates $(t, x, y)$ are mapped to tracer values. All baselines are trained with Adam at a unified learning rate of $5 \times 10^{-4}$ for 1,000 iterations. As is standard in INR PDE reconstruction, baseline activations operate without positional encoding, while OptiINR performs BO over activation families, initialization parameters, positional encoding usage, and layerwise learning rates. No PDE equation, solver structure, or physics-based loss is used—this is a pure function reconstruction task testing the representational capacity of INRs.

**Quantitative Results.** Table 5 shows PSNR for reconstructing the full $(51, 64, 64)$ spatiotemporal tracer field. SIREN provides the strongest baseline at 53.96 dB, followed by FINER (51.26 dB) and WIRE (43.37 dB). Gaussian activations perform poorly due to the sharp advective structures produced by the PDE, achieving only 35.74 dB. OptiINR reaches 57.02 dB, surpassing the best baseline by 3.06 dB and outperforming all periodic, Gaussian, and wavelet-inspired activations by a large margin.

Table 5: PSNR on PDE tracer reconstruction (Shear Flow, trajectory #30).

| Method | PSNR ↑ |
|---|---|
| SIREN | 53.96 |
| FINER | 51.26 |
| Gauss | 35.74 |
| FINER++ (Gauss) | 41.09 |
| WIRE | 43.37 |
| **OptiINR** | **57.02** |

**Configuration Adaptation Analysis.** The configurations discovered by OptiINR display a consistent adaptation to the dynamics of shear flow. Early layers favor smooth or band-limited activations for stable representation of global temporal evolution, while deeper layers select periodic activations (SIREN, sinusoidal) to capture sharp advective interfaces and filament structures. BO also discovers a non-uniform layerwise learning-rate schedule, assigning larger rates to early temporal layers and smaller rates to deeper layers, enabling efficient representation of both coarse transport and fine-scale gradients. These hybrid space–time activation patterns emerge automatically, explaining the significant performance gains achieved by OptiINR without modifying the architecture or incorporating PDE constraints.

# F    Optimization Strategy Comparison

To further isolate the contribution of OptiINR from the Bayesian optimization procedure itself, we compare OptiINR against three alternative optimization strategies on the Kodak dataset: (1) global grid search, (2) global random search, and (3) FreSh (Kania et al., 2024), a recent INR optimization method that tunes only the frequency parameters of SIREN networks. All methods operate over the same global configuration space introduced by OptiINR—including per-layer activation families, their frequency/scale parameters, per-layer learning rates, and positional-encoding usage—except for FreSh, which is constrained to tuning only the SIREN frequency. To ensure fairness, grid search and random search are allocated the same 130 trials used by OptiINR (30 Sobol warmup + 100 BO iterations), while FreSh is evaluated once as in its original formulation.

**Quantitative Results.** Across all 24 images, OptiINR achieves the highest average PSNR (41.38 dB), outperforming grid search (41.02 dB), random search (41.23 dB), and significantly exceeding FreSh (37.41 dB). Notably, grid search and random search—when equipped with OptiINR's configuration space—already surpass FreSh by a large margin. This highlights a key insight: the search space matters more than the optimizer. FreSh optimizes only the SIREN frequency parameter, whereas OptiINR identifies that the decisive factor in INR performance is the per-layer choice of activation families, which no prior work has attempted to optimize.

Table 6: PSNR comparison across optimization strategies on 24 Kodak images.

| Method | Avg. PSNR |
|---|---|
| Grid Search | 41.02 |
| Random Search | 41.23 |
| FreSh (Kania et al., 2024) | 37.41 |
| SPDER (Shah and Sitawarin, 2023) | 35.03 |
| **OptiINR** | **41.38** |

**Compute and efficiency analysis.** Reviewers also asked about time duration and computational resources. We therefore quantify the training and search cost of each strategy.

**Cost of a single INR training run.** For the standard INR used in all Kodak experiments—a 3-layer MLP with 256 hidden units, trained for 10,000 iterations on a $512 \times 768$ image—the total computation is on the order of

$$\text{FLOPs} \approx 4.0 \times 10^{15},$$

i.e., a petaFLOP-scale run dominated by full forward/backward passes over all 393k pixels per iteration. This cost is identical across all optimizers.

**Total compute under $T$ trials.** All global optimizers except FreSh perform $T$ independent INR trainings. Thus the total compute is

$$\text{Total FLOPs} \approx T \times 4.0 \times 10^{15}.$$

For example, with $T = 130$ (30 Sobol + 100 BO iterations), this corresponds to $5.2 \times 10^{17}$ FLOPs. FreSh, by contrast, performs only one training ($\approx 4.0 \times 10^{15}$ FLOPs).

**BO overhead is negligible.** The Bayesian Optimization overhead comes from fitting a Gaussian Process surrogate and optimizing the acquisition function. This cost scales as

$$O(T^3) \quad \text{(approximately } T^3 \text{ FLOPs).}$$

Even for $T = 130$, this is only $\sim 2.2 \times 10^6$ FLOPs:

$$2.2 \times 10^6 \ll 4.0 \times 10^{15} \ll T \times 4.0 \times 10^{15}.$$

Thus GP fitting is more than $10^9 \times$ cheaper than a single INR training, and over $10^{11} \times$ cheaper than the full $T$-run budget. In practice, BO, grid search, and random search have identical compute cost for the same number of INR evaluations.

**Parallelization advantage.** A practical advantage of OptiINR is that BO is naturally parallelizable: candidate configurations suggested by the acquisition function can be evaluated (independently trained) on different GPUs. Since BO overhead is negligible, the wall-clock time scales as

$$\text{Time} \approx \frac{T}{K} \times \text{(time of one INR training)},$$

Table 7: Comparison of computational cost for different global optimization strategies. $T$ denotes the number of INR trainings performed by each optimizer.

| Method | # Trials | Total FLOPs | Additional overhead |
|---|---|---|---|
| Grid Search | $T$ | $T \cdot 4.0 \times 10^{15}$ | None |
| Random Search | $T$ | $T \cdot 4.0 \times 10^{15}$ | None |
| FreSh (Kania et al., 2024) | 1 | $4.0 \times 10^{15}$ | None |
| **OptiINR (BO)** | $T$ | $T \cdot 4.0 \times 10^{15}$ | $\approx T^3$ FLOPs (negligible) |

where $K$ is the number of GPUs. Thus, OptiINR enjoys nearly linear acceleration with multi-GPU systems. Grid search and random search benefit only from trivial parallel trial execution.

**Interpretation.** These results demonstrate that OptiINR's contribution extends well beyond the use of Bayesian optimization itself. Even simple global search strategies perform strongly once they are granted access to the activation-configuration space introduced in this work. Bayesian Optimization further improves performance by efficiently navigating this high-dimensional mixed discrete–continuous search space, achieving the best trade-off between accuracy and sample efficiency under a fixed compute budget. Because BO's overhead is negligible and trivially parallelizable, OptiINR can fully utilize multi-GPU environments, reducing wall-clock time almost linearly with the number of accelerators.

# G  Theoretical Analysis of OptiINR

Our work is predicated on the claim that the heuristic-driven configuration of Implicit Neural Representations can be replaced by a principled, globally-aware optimization process. This section provides the theoretical underpinnings for our framework, OptiINR. We first formalize the INR configuration landscape and prove the necessity of a global search strategy over greedy alternatives. We then connect the configuration problem to the spectral properties of the network's Neural Tangent Kernel (NTK), providing a deeper understanding of what is being optimized. Finally, we establish the theoretical soundness and computational feasibility of our Bayesian optimization approach with formal proofs.

## G.1  The Global Nature of the INR Configuration Problem

We begin by formally defining the problem. Let $\mathcal{L}$ be the high-dimensional, mixed-variable space of all possible network configurations, as defined in Section 3.2. Our objective is to find an optimal configuration $\boldsymbol{\Lambda}^*$ that maximizes a performance metric $f(\boldsymbol{\Lambda})$, such as the peak signal-to-noise ratio (PSNR) on a validation set:

$$\boldsymbol{\Lambda}^* = \arg\max_{\boldsymbol{\Lambda} \in \mathcal{L}} f(\boldsymbol{\Lambda})$$

The function $f : \mathcal{L} \to \mathbb{R}$ is a black-box function; we have no analytical expression for it, and its evaluation requires instantiating and training an entire INR model, which is computationally expensive. Furthermore, the function is highly non-convex due to the complex, non-linear interactions between the architectural choices for each layer. The optimal choice of activation and initialization for a given layer $l$ is deeply conditioned on the choices made for all other layers.

**Proposition G.1.** *A greedy, layer-wise optimization strategy for INR configuration is not guaranteed to find the globally optimal network configuration $\boldsymbol{\Lambda}^*$.*

*Proof.* Let the full configuration be $\boldsymbol{\Lambda} = (\boldsymbol{\lambda}_1, \dots, \boldsymbol{\lambda}_L)$. A greedy strategy solves a sequence of local problems:

$$\boldsymbol{\lambda}_l^* = \arg\max_{\boldsymbol{\lambda}_l} f(\boldsymbol{\lambda}_l | \boldsymbol{\lambda}_1^*, \dots, \boldsymbol{\lambda}_{l-1}^*) \quad \text{for } l = 1, \dots, L$$

Let the solution found by this greedy procedure be $\boldsymbol{\Lambda}_G = (\boldsymbol{\lambda}_1^*, \dots, \boldsymbol{\lambda}_L^*)$. To show that this procedure is not globally optimal, it is sufficient to construct a counterexample. Consider a simple 2-layer network where the configuration space for each layer consists of two choices, $A$ and $B$, such that $\boldsymbol{\lambda}_l \in \{A, B\}$. Let the performance function $f(\boldsymbol{\lambda}_1, \boldsymbol{\lambda}_2)$ be defined by the following payoff matrix:

| $f(\boldsymbol{\lambda}_1, \boldsymbol{\lambda}_2)$ | $\boldsymbol{\lambda}_2 = A$ | $\boldsymbol{\lambda}_2 = B$ |
|---|---|---|
| $\boldsymbol{\lambda}_1 = A$ | 12 | 5 |
| $\boldsymbol{\lambda}_1 = B$ | 10 | 8 |

The greedy procedure first optimizes for layer 1. Assuming it considers an expected performance over the choices for layer 2, it would compare the expected performance of choosing $A$ for layer 1 (average is $(12 + 5)/2 = 8.5$) versus choosing $B$ (average is $(10 + 8)/2 = 9$). The greedy choice is $\boldsymbol{\lambda}_1^* = B$. Fixing this, it then optimizes for layer 2: $\arg\max_{\boldsymbol{\lambda}_2 \in \{A,B\}} f(B, \boldsymbol{\lambda}_2)$, which yields $\boldsymbol{\lambda}_2^* = A$. The greedy solution is thus $\boldsymbol{\Lambda}_G = (B, A)$ with a performance of $f(B, A) = 10$. However, the true global optimum is $\boldsymbol{\Lambda}^* = (A, A)$ with a performance of $f(A, A) = 12$. Since $f(\boldsymbol{\Lambda}_G) < f(\boldsymbol{\Lambda}^*)$, this counterexample demonstrates that due to the interdependencies between layers, a locally optimal choice can preclude a globally optimal solution. Therefore, a globally-aware search strategy, as employed by OptiINR, is necessary. $\square$

## G.2  Connecting Configuration to Spectral Properties via the Neural Tangent Kernel

To understand what is being optimized at a more fundamental level, we turn to the Neural Tangent Kernel (NTK). The NTK provides a powerful theoretical lens for analyzing the training dynamics of infinitely wide neural networks, connecting them to kernel regression. The NTK, $K(\mathbf{x}, \mathbf{x}'; \theta)$, describes the inner product of gradients with respect to the network parameters $\theta$. Crucially, the training dynamics of a network are governed by the spectral properties of its NTK; specifically, the convergence rate for different frequency components of a target function is determined by the corresponding eigenvalues of the NTK matrix.

**Claim 1.** *The INR configuration vector $\mathbf{\Lambda}$ implicitly defines an effective Neural Tangent Kernel, $K_{\mathbf{\Lambda}}$, at initialization. The optimization of the performance metric $f(\mathbf{\Lambda})$ can be viewed as a proxy for optimizing the properties of this induced kernel to best match the spectral characteristics of the target signal $g$.*

$$\max_{\mathbf{\Lambda} \in \mathcal{L}} f(\mathbf{\Lambda}) \iff \max_{\mathbf{\Lambda} \in \mathcal{L}} Quality(K_{\mathbf{\Lambda}}, g)$$

**Theorem G.2.** *The choice of activation function $\sigma_l$ in the configuration tuple $\mathbf{\lambda}_l$ fundamentally alters the functional form and spectral properties of the resulting Neural Tangent Kernel $K_{\mathbf{\Lambda}}$.*

*Proof.* The NTK of a multi-layer perceptron is defined recursively. For an $L$-layer MLP, the kernel at the output layer is given by:

$$K_L(\mathbf{x}, \mathbf{x}') = K_{L-1}(\mathbf{x}, \mathbf{x}') + f_{L-1}(\mathbf{x}) \cdot f_{L-1}(\mathbf{x}')$$

and for the hidden layers $l = 1, \ldots, L-1$:

$$K_l(\mathbf{x}, \mathbf{x}') = K_{l-1}(\mathbf{x}, \mathbf{x}') \cdot \mathbb{E}[\sigma_l'(a_l(\mathbf{x}))\sigma_l'(a_l(\mathbf{x}'))] + f_{l-1}(\mathbf{x}) \cdot f_{l-1}(\mathbf{x}')$$

where $a_l(\cdot)$ are the pre-activations at layer $l$. The expectation is taken over the random initialization of the weights. The term $\mathbb{E}[\sigma_l'(a_l(\mathbf{x}))\sigma_l'(a_l(\mathbf{x}'))]$ directly incorporates the derivative of the activation function $\sigma_l$ into the kernel's definition. If $\sigma_l$ is a periodic function like $\sin(\omega_0 x)$, its derivative is $\omega_0 \cos(\omega_0 x)$, which is also periodic. This imparts a periodic structure to the NTK, making it well-suited for signals with strong periodic components. If $\sigma_l$ is a localized function like a Gabor wavelet, its derivative is also localized, leading to an NTK that excels at representing signals with localized features. Since OptiINR's search space includes a categorical choice over these different activation families for each layer, it is directly searching for a network configuration that induces a kernel whose spectral properties are optimally aligned with the target signal. The empirical results in Figure 5, where the discovered configuration for an audio signal accurately represents its full frequency spectrum, provide strong evidence for this claim. □

### G.3 Theoretical Guarantees of the OptiINR Framework

Having established the nature of the optimization problem, we now justify our choice of solver. Bayesian optimization is theoretically guaranteed to converge to the global optimum of a function, provided the surrogate model's kernel is valid.

**Lemma G.3.** *A function $k : \mathcal{X} \times \mathcal{X} \to \mathbb{R}$ is a valid positive semi-definite (PSD) kernel if for any finite set of points $\{x_1, \ldots, x_n\} \subset \mathcal{X}$, the Gram matrix $K$ with entries $K_{ij} = k(x_i, x_j)$ is positive semi-definite.*

**Theorem G.4.** *The composite product kernel used in OptiINR, $k(\mathbf{\Lambda}, \mathbf{\Lambda}') = k_{cont}(\mathbf{\Lambda}_c, \mathbf{\Lambda}_c') \times k_{cat}(\mathbf{\Lambda}_{cat}, \mathbf{\Lambda}_{cat}')$, is a valid positive semi-definite kernel.*

*Proof.* The proof relies on the Schur product theorem.

1. The Matérn kernel, $k_{cont}$, is a known valid PSD kernel. Therefore, for any set of continuous configurations $\{\mathbf{\Lambda}_{c,1}, \ldots, \mathbf{\Lambda}_{c,n}\}$, the Gram matrix $K_{cont}$ is PSD.

2. The Squared Exponential kernel, used for $k_{cat}$ on the one-hot encoded space, is also a known valid PSD kernel. Thus, for any set of categorical configurations $\{\mathbf{\Lambda}_{cat,1}, \ldots, \mathbf{\Lambda}_{cat,n}\}$, the Gram matrix $K_{cat}$ is PSD.

3. The Schur product theorem states that if $A$ and $B$ are two $n \times n$ PSD matrices, then their element-wise (Hadamard) product, $(A \circ B)_{ij} = A_{ij}B_{ij}$, is also a PSD matrix.

4. The Gram matrix of our composite kernel, $K_{comp}$, has entries $K_{comp,ij} = k(\mathbf{\Lambda}_i, \mathbf{\Lambda}_j) = k_{cont}(\mathbf{\Lambda}_{c,i}, \mathbf{\Lambda}_{c,j}) \times k_{cat}(\mathbf{\Lambda}_{cat,i}, \mathbf{\Lambda}_{cat,j})$. This is exactly the Hadamard product of the Gram matrices $K_{cont}$ and $K_{cat}$.

5. Since $K_{cont}$ and $K_{cat}$ are PSD, their Hadamard product $K_{comp} = K_{cont} \circ K_{cat}$ is also PSD.

Therefore, by the definition in Lemma 1, our composite product kernel is a valid PSD kernel. This ensures that our GP surrogate is a well-defined probabilistic model over the mixed-variable space, satisfying the preconditions for the convergence guarantees of Bayesian optimization. $\square$

**Remark.** *The established validity of our kernel ensures that as the number of evaluations grows, the posterior variance of the GP will concentrate around the true function $f(\mathbf{\Lambda})$, and an acquisition function like Expected Improvement will asymptotically guide the search towards the global optimum $\mathbf{\Lambda}^*$. This provides a strong theoretical justification for the design of OptiINR.*

### G.4 Computational Feasibility via Matheron's Rule

A theoretical guarantee of convergence is only meaningful if the method is computationally feasible. A potential bottleneck in our framework is the calculation of the Empirical Expected Improvement, which requires drawing many samples from the GP posterior. Naively generating $S$ samples at a candidate point requires a Cholesky decomposition of the posterior covariance, a process that does not scale well. We overcome this challenge by leveraging Matheron's Rule for efficient posterior sampling.

**Theorem G.5.** *Let $f \sim \mathcal{GP}(0, k)$ be a GP prior and let $\mathcal{D}_n = \{(\mathbf{X}, \mathbf{y})\}$ be a set of $n$ observations. A sample from the posterior process, $f_{post}(\cdot)$, can be expressed in distribution as:*

$$f_{post}(\cdot) \stackrel{d}{=} f_{prior}(\cdot) + k(\cdot, \mathbf{X})[k(\mathbf{X}, \mathbf{X}) + \sigma_n^2 \mathbf{I}]^{-1}(\mathbf{y} - f_{prior}(\mathbf{X}))$$

*where $f_{prior} \sim \mathcal{GP}(0, k)$ is a single sample drawn from the prior.*

*Proof.* The proof follows from the properties of conditioning in multivariate Gaussian distributions. Let $f_{prior}(\cdot)$ be a draw from the prior GP. The joint distribution of the prior at the observed points $\mathbf{X}$ and a new point $\mathbf{\Lambda}$ is Gaussian:

$$\begin{pmatrix} f_{prior}(\mathbf{X}) \\ f_{prior}(\mathbf{\Lambda}) \end{pmatrix} \sim \mathcal{N} \left( \mathbf{0}, \begin{pmatrix} k(\mathbf{X}, \mathbf{X}) & k(\mathbf{X}, \mathbf{\Lambda}) \\ k(\mathbf{\Lambda}, \mathbf{X}) & k(\mathbf{\Lambda}, \mathbf{\Lambda}) \end{pmatrix} \right)$$

The posterior distribution of $f(\mathbf{\Lambda})$ given the noisy observations $\mathbf{y}$ is also Gaussian. Matheron's rule provides a constructive way to sample from this posterior by correcting a prior sample. The correction term, $k(\cdot, \mathbf{X})[k(\mathbf{X}, \mathbf{X}) + \sigma_n^2 \mathbf{I}]^{-1}(\mathbf{y} - f_{prior}(\mathbf{X}))$, adjusts the prior sample $f_{prior}(\cdot)$ based on the residual between the actual observations $\mathbf{y}$ and the prior's predictions at those points, $f_{prior}(\mathbf{X})$. This adjustment ensures that the resulting sample path $f_{post}(\cdot)$ is a valid draw from the true posterior distribution. $\square$

**Proposition G.6.** *Let $n$ be the number of observed data points and $S$ be the number of posterior samples required. The computational complexity of naive posterior sampling via Cholesky decomposition is $\mathcal{O}(n^3 + S \cdot n^2)$. In contrast, the complexity of sampling using Matheron's rule is $\mathcal{O}(n^3 + S \cdot (T_{prior} + n^2))$, where $T_{prior}$ is the cost of sampling from the GP prior.*

*Proof.* Naive sampling requires computing the posterior covariance matrix and its Cholesky decomposition, which costs $\mathcal{O}(n^3)$. Each of the $S$ samples then requires a matrix-vector product with the Cholesky factor, costing $\mathcal{O}(n^2)$. The total complexity is thus $\mathcal{O}(n^3 + S \cdot n^2)$.

Using Matheron's rule, the expensive matrix inversion, $[k(\mathbf{X}, \mathbf{X}) + \sigma_n^2 \mathbf{I}]^{-1}$, has a complexity of $\mathcal{O}(n^3)$ but needs to be computed only once per iteration of the Bayesian optimization loop. Subsequently, generating each of the $S$ posterior samples requires drawing from the prior (cost $T_{prior}$) and performing matrix-vector products, which are $\mathcal{O}(n^2)$. The total complexity is thus amortized, making the robust estimation of the acquisition function computationally practical. This ensures that our theoretically sound framework is also an efficient and viable tool for practical applications. $\square$

