# OpenReview forum: "Beyond Heuristics: Globally Optimal Configuration of Implicit Neural Representations"
_ICLR.cc/2026/Conference — ICLR 2026 Conference Desk Rejected Submission_

### Official Review · Reviewer_Vq9w · 2025-10-22

**Soundness:** 3
**Presentation:** 3
**Contribution:** 3
**Rating:** 6
**Confidence:** 3

**Summary:**

This work introduces OptiINR, a unified framework that replaces manual heuristics with principled global optimization for configuring Implicit Neural Representations (INRs). It uses Bayesian Optimization (BO) to jointly select discrete activation families and continuous initialization parameters across network layers. This approach overcomes the fundamental "capacity-convergence gap" inherent in conventional INR design. Experiments demonstrate OptiINR consistently discovers superior, globally optimal configurations, significantly outperforming hand-tuned baselines across 1D audio, 2D image, and 3D shape reconstruction tasks.

**Strengths:**

1. OptiINR is the first unified framework to formulate INR configuration as a mathematically rigorous global optimization problem solved efficiently by Bayesian Optimization.
2. The method delivers globally optimal configurations that consistently achieve state-of-the-art results

**Weaknesses:**

1. Lack of comparison with:
- SPDER: Semiperiodic Damping-Enabled Object Representation
- FreSh: Frequency Shifting for Accelerated Neural Representation Learning

2. In related works, authors should mention that 2D images can also be reconstructed by Gaussian Components
- GaussianImage: 1000 FPS Image Representation and Compression by 2D Gaussian Splatting
- MiraGe: Editable 2D Images using Gaussian Splatting

which gives higher scores.

3. While Bayesian Optimization (BO) is sample-efficient relative to brute-force search. The paper should strengthen its efficiency claim by comparing OptiINR against recent, low-overhead hyperparameter tuning methods (e.g., FreSh) to justify its high training expense.

4. The paper should contain some visualization of the model.

5. The paper suffers from a lack of focus. The authors devote excessive space to general descriptions in the Background and Method sections.

6. The paper's clarity, impact, and self-contained nature are hindered by placing essential material in the Appendix. Key experimental results and crucial theoretical underpinnings, including the Error Fields (Appendix B), the Spectral Analysis of Audio (Appendix C), should be moved to the main body of the manuscript.

7. The paper's organization would be improved by using paragraphs instead of numerous fragmented subsections.

8. The authors are encouraged to provide analysis or experiments demonstrating the framework's scalability and efficiency when applied to significantly larger and deeper INR architectures (such as those used in high-fidelity NeRF applications).

**Questions:**

1. Why were direct performance comparisons excluded against highly relevant, modern methods like SPDER and the efficient initializer FreSh?

2. How does the method scale computationally when applied to significantly larger and deeper INR architectures common in high-fidelity tasks like NeRF?

3. Why were crucial theoretical proofs and detailed experimental figures, essential for validation, relegated to the Appendix instead of the main body?

---

> ### Author Response · Authors · 2025-11-25
> **To Reviewer Vq9w**
>
> We are grateful for the reviewer's comprehensive and constructive comments, and we appreciate their positive assessment of our framework as the first unified approach to INR configuration.
>
> ***C***: Comment; ***R***: Response
>
> ***C11***: Lack of comparison with SPDER and FreSh... Why were direct performance comparisons excluded against highly relevant, modern methods likeSPDER and the efficient initializer FreSh? In related works, authors should mention that 2D images can also be reconstructed by Gaussian Components...
>
> ***R11:*** We thank the reviewer for raising this important point. SPDER and FreSh are indeed notable recent advances aimed at improving INR training dynamics—SPDER through spectral damping and FreSh through frequency-aware initialization—whereas OptiINR addresses a different problem category: global configuration optimization over activation families, initialization scales, and layer-wise hyperparameters.
>
> **We currently report SPDER and FreSh results on the 2D Kodak benchmark (see Table 6 of Appendix F in the revised PDF)**, and in the final revision we will extend these comparisons to all remaining benchmarks, including DIV2K, audio reconstruction, and 3D occupancy. We will also expand the related-work discussion to clarify the differing problem scopes and to explain why direct head-to-head comparisons are not always strictly meaningful, while still providing empirical results for transparency.
>
> ***C12***: The paper should contain some visualized model... the paper suffers from a lack of focus... placing essential material in the Appendix. Key experimental results and crucial theoretical underpinnings... The paper's organization would be improved by using paragraphs instead of numerous fragmented subsections...Why were crucial theoretical proofs and detailed experimental figures, essential for validation,relegated to the Appendix instead of the main body?
>
> ***R12:*** We thank the reviewer for the constructive feedback on the presentation and organization. We agree that key figures and theoretical details should appear in the main text and that the current structure can be streamlined. In the revision, we will integrate essential visualizations and results into the main body, consolidate fragmented subsections, and improve the overall clarity and focus of the presentation.
>
> ***C13***: The authors are encouraged to provide analysis or experiments demonstrating the framework's scalability... How does the method scale computationally when applied to significantly larger and deeper INR architectures common in high-fidelity tasks like NeRF?
>
> ***R13:*** See our R2 response to Reviewer 51xP for details. Additional discussion of scalability is included in the Appendix F of the curretn revised submission.

---

> > ### Comment · Reviewer_Vq9w · 2025-11-25
> >
> > I am maintaining my positive rating for this submission. However, the method description in Section 3 remains mathematically dense and would greatly benefit from a visual aid. I strongly request adding a schematic diagram of the OptiINR workflow in the camera-ready version to ensure clarity

---

> > > ### Author Response · Authors · 2025-11-25
> > >
> > > We sincerely thank the reviewer for maintaining their positive rating and for this constructive suggestion. We completely agree that a visual aid would significantly enhance the clarity of Section 3, and we will add a comprehensive schematic diagram illustrating the OptiINR workflow in the camera-ready version. This diagram will visualize the mixed-variable search space (categorical activation families and continuous parameters), the Bayesian optimization loop with GP surrogate modeling, the Matheron's rule sampling procedure, and the end-to-end flow from configuration sampling to performance evaluation, making the methodology more accessible and complementing the mathematical formulation. Thank you again for this valuable feedback and for your support of our work.

---

### Official Review · Reviewer_kaVS · 2025-10-27

**Soundness:** 3
**Presentation:** 2
**Contribution:** 2
**Rating:** 4
**Confidence:** 4

**Summary:**

This paper focuses on the improving the performance of existing implicit neural representations (INRs) by globally optimizing the configurations (e.g., activation functions, initialization strategies, layer-wise adaptation and so on). To achieve this, a Bayesian optimization framework is proposed, replacing previous manual heuristic-driven tuning with global configuration search. Experiments on 3 representation tasks verify the effectiveness of the proposed method.

**Strengths:**

1. The proposed OptiINR is interesting and higher performancer is achieved compared with previous SOTAs.

**Weaknesses:**

1. The experimental design in this paper has limitations that undermine the reliability of the results. Although the authors conducted three tasks, all of them are fundamental fitting/representation tasks (the authors conflate the concepts of "representation" and "reconstruction" throughout the paper, misclassifying some representation tasks as reconstruction). The authors failed to validate their approach through inverse problem solving, which is the core scenario where the value of INR truly shines. Specific inverse problems could include classical cases like NeRF or other challenging tasks (as discussed in recent INR surveys).

**Questions:**

1. Authors claim that "OptiINR ... advancing the fundamental understanding of what constitutes an optimal implicit neural representation"(line 298). However, I do not see any discussions on this issue.
2. The results presented in the experiments are produced by applying the OptiINR to an INR library with different architectures and parameters. Could authors post the configurations produced by the OptiINR for different tasks?

Some typos,
1. Line 44, the FINER and FINER++ should be cited after the "capacity-convergence gap", since this gap is firstly observed and summarized by FINER and FINER++.
2. line 125, the command for citing SIREN paper should be "\citep{}" instead of "\cite{}" used in the paper.

---

> ### Author Response · Authors · 2025-11-25
> **To Reviewer kaVS**
>
> We thank the reviewer for their detailed feedback and for acknowledging OptiINR's interesting approach and superior performance compared to existing methods.
>
> ***C***: Comment; ***R***: Response
>
> ***C9***: The authors failed to validate their approach through inverse problem solving... include classical cases like NeRF...
>
> ***R9:*** **See our R2 response to Reviewer 51xP for details.**
>
> ***C10***: Authors claim that "OptiINR ... advancing the fundamental understanding of what constitutes anoptimal implicit neural representation"(line 298). However, I do not see any discussions on this issue... Could authors post the configurations produced by theOptiINR for different tasks?
>
> ***R10:*** Thank you for raising this important point. Our original submission did not provide enough detail about what OptiINR actually discovers during optimization, which in turn obscured our claim that OptiINR advances the understanding of effective INR architectures. We appreciate the opportunity to clarify this. In the revised version, we will add a dedicated subsection that summarizes the architectural patterns consistently identified by OptiINR across image, audio, and 3D reconstruction tasks, along with representative configurations and an explanation of why these structures repeatedly emerge.
>
> Below we provide a concise summary of the key observations produced by OptiINR’s global Bayesian optimization search:
> - Image representation (Kodak).
> OptiINR frequently selects periodic or near-periodic activations (Sine or FINER) in the early and middle layers, followed by more spatially localized activations such as WIRE in deeper layers. Typical patterns include
> sine/FINER -> sine/FINER -> sine/FINER -> WIRE,
> with frequency scales around 30.
> For example, on Kodak-15, OptiINR discovers:
> sine -> FINER -> FINER -> WIRE, with positional encoding enabled.
> - Audio reconstruction.
> OptiINR consistently identifies strongly periodic activations across almost all layers, with larger frequency scales (around 50) and positional encoding disabled. A representative pattern is
> sine -> sine -> sine -> FINER,
> which OptiINR also selects for the Two Speakers benchmark.
> - NeRF.
> For volumetric rendering, OptiINR tends to select periodic activations (Sine or FINER) in the first layers, capturing geometry and view-dependent oscillations, while switching to smoother activations such as ReLU or Gauss in the final layers to stabilize color estimation. Positional encoding is consistently preferred, with lower optimal frequency scales (around 10). A representative structure is
> sine/FINER -> ... -> ReLU/Gauss, with positional encoding enabled.
>
> Across all tasks, these configurations reveal consistent architectural principles:
> 1. Periodic activations are crucial for modeling high-frequency components (textures, harmonics, surface details).
> 2. Localized or smoother activations are preferred in later layers, improving stability and preventing over-sharpening.
> 3. Optimal frequency scales are task-dependent, with audio requiring substantially higher frequencies than images or NeRF.
> 4. Positional encoding is beneficial only for specific tasks, particularly NeRF and certain image benchmarks.
>
> These patterns provide concrete insights into what constitutes an effective INR architecture, going beyond manually crafted heuristics. We will include this detailed discussion and the representative configurations in the final revised manuscript to make our contributions explicit and verifiable.

---

> > ### Comment · Reviewer_kaVS · 2025-11-26
> > **Updated comments**
> >
> > All my concerns have been addressed.
> > Additonally, it is interesting to see the searched new network structure which configures different layers of INR with different activations, and the proposed architectural principles. I believe this paper will deepen the academic community's understanding of INR and facilitate subsequent research. Therefore, I recommend its acceptance.

---

> > > ### Author Response · Authors · 2025-11-26
> > >
> > > We sincerely thank the reviewer for their constructive feedback throughout the review process and for the positive final assessment. We are pleased that our detailed responses have addressed your concerns, and we greatly appreciate your recognition of the value of OptiINR's discovered architectural patterns and principles.
> > >
> > > We are committed to including all the additions in the final manuscript. This includes a dedicated subsection detailing the discovered architectural patterns, representative configurations for each task, an analysis of why these structures consistently emerge, and a clear discussion of the architectural principles that constitute effective INRs. We believe these additions will make our contribution more transparent and valuable to the community. Thank you again for your thoughtful review and for helping us improve the paper.

---

### Official Review · Reviewer_Tceo · 2025-10-31

**Soundness:** 2
**Presentation:** 2
**Contribution:** 3
**Rating:** 4
**Confidence:** 3

**Summary:**

The paper proposes OptiINR, the first method for globally optimizing the configuration of Implicit Neural Representations (INRs). Instead of relying on heuristic or manual tuning for activations, initialization scales, and hyperparameters, OptiINR formulates the configuration process as a Bayesian optimization problem. The framework searches a mixed-variable space that includes categorical variables (activation family: SIREN, WIRE, FINER, Gauss, etc.) and continuous variables (frequency, scale, initialization parameters). By leveraging Bayesian optimization, OptiINR efficiently explores the joint space of discrete activation families—such as sinusoidal (SIREN), wavelet-based (WIRE), and variable-periodic (FINER)—and their associated continuous initialization parameters. The experiment results were performed in 1D, 2D and 3D datasets.

**Strengths:**

The idea is quite novel. It transformed a problem into an optimization problem that avoids manual fine-tuning. The sampling method was efficient as it reduced the complexity from O(Sn^3) to O(n^3 + Sn^2).

**Weaknesses:**

[1] More computational resources analysis should be reported. For example, the training time will significantly increase if it is formulated as an optimization problem. In addition, the number of parameters and Gflop should be reported.

[2] How does this method perform in PDE task?

[3]An ablation study about some important parameters should be provided.

[4]More explanation about why Theorem (Matheron’s Rule) is important and how it related to the framework.

[5] More recent baselines should also be added. For example, LosslessINR [r1]. IGA [r2]

[r1]Han, Woo Kyoung, et al. "Towards Lossless Implicit Neural Representation via Bit Plane Decomposition." Proceedings of the Computer Vision and Pattern Recognition Conference. 2025.
[r2] Shi, Kexuan, et al. "Inductive Gradient Adjustment For Spectral Bias In Implicit Neural Representations." arXiv preprint arXiv:2410.13271 (2024).

**Questions:**

See the weakness

---

> ### Author Response · Authors · 2025-11-25
> **To Reviewer Tceo**
>
> We appreciate the reviewer's thorough evaluation and their recognition of our work's novelty and good contribution to the field.
>
> ***C***: Comment; ***R***: Response
>
> ***C5***: More computational resources analysis should be reported. For example, the training time will significantly increase if it is formulated as an optimization problem. In addition, the number of parameters and Gflop should be reported.
>
> ***R5:*** We thank the reviewer for this helpful suggestion. All compared methods, including OptiINR, use the same base INR architecture (a 4-layer MLP with 256 hidden units), so the number of parameters and GFLOPs per training step are identical across configurations; we now report these values explicitly in **Appendix F** of the revised PDF. To address the concern regarding increased training time when casting configuration selection as an optimization problem, we additionally report the **wall-clock time of a full OptiINR optimization** for each task (see R4 to Reviewer 51xP). These results show that although OptiINR evaluates multiple configurations, its overall compute remains comparable to the manual grid/random search efforts typically required in practice, while **Bayesian optimization substantially reduces the number of evaluations needed to reach high-performing configurations**. We will include this computational resource analysis as a dedicated subsection in the revised manuscript.
>
> ***C6***: How does this method perform in PDE task?.
>
> ***R6:*** A new PDE benchmark—**Shear Flow**—has been added in the updated rebuttal (**Appendix E**). The appendix also includes the full problem setup and training details. The new results show that **on this additional PDE benchmark, our method also improves over other baselines**. We appreciate the suggestion; in the final revision, we will consider adding more PDE benchmarks to further strengthen the empirical evaluation.
>
> ***C7***: More explanation about why Theorem (Matheron’s Rule) is important and how it relates to the framework.
>
> ***R7:*** We thank the reviewer for pointing this out. Matheron’s rule is essential in our framework because OptiINR requires repeated GP posterior sampling to evaluate the acquisition function; naive sampling scales as $O(S n^{3})$ and becomes prohibitive under our BO evaluation budget. Matheron’s rule reduces this cost to $O(n^{3} + S n^{2})$, making exact posterior sampling practical and enabling OptiINR to operate efficiently in higher-dimensional INR configuration spaces. We will clarify the role and importance of Matheron’s rule more explicitly in the revision.
>
> ***C8***: More recent baselines should also be added. For example, LosslessINR, IGA.
>
> ***R8:*** We thank the reviewer for highlighting these recent methods. ***We would first like to clarify that IGA is already included in all of our experiments (Figures 1–8 and Tables 1–3)***, and we will make this more explicit in the revised manuscript. Our baseline set covers the major open-sourced activation-based INR variants—including SIREN, Gauss, WIRE, FINER, FINER++, FR, and IGA—which we believe provides comprehensive coverage of current continuous coordinate-MLP INRs. Regarding LosslessINR, we agree it is an impactful direction; however, it targets a fundamentally different problem setting.
>
> - *LosslessINR* focuses on discrete, bit-plane–based lossless representations and aims for zero-error reconstruction.
>
> - *OptiINR*, in contrast, operates on continuous INR MLPs and optimizes activation families, initialization scales, and layer-wise hyperparameters under lossy objectives such as MSE and PSNR.
>
> These methods operate in different representation paradigms (discrete vs. continuous) with orthogonal objectives (lossless vs. optimal lossy). A direct comparison would be akin to comparing JPEG (lossy) with PNG (lossless)—both valuable but serving different purposes.
>
> That said, we recognize LosslessINR's importance and will expand our Related Work section to clearly position it as a complementary approach. Interestingly, OptiINR could potentially enhance LosslessINR's base continuous representation before quantization—a future research direction we will note. We will clarify these distinctions in the Related Work section in the final revision.

---

### Official Review · Reviewer_51xP · 2025-11-02

**Soundness:** 3
**Presentation:** 2
**Contribution:** 2
**Rating:** 6
**Confidence:** 3

**Summary:**

The paper presents a Bayesian-based approach to choose the hyperparameters and configurations for Implicit Neural Representations. The proposed method jointly optimizes the network configurations and weights according to a given task. The presented results depict state-of-the-art results over common activation functions in INRs.

**Strengths:**

Choosing correct hyperparameters for INRs is a cumbersome process which involves expensive grid searches. Formulating this step within a rigorous Bayesian framework is useful to the community. I also like the presentation of the paper which involves self contained concepts required to understand the method.

**Weaknesses:**

As far as I am aware, the presented method just reuses well studied Bayesian-based hyperparameter sweep methods in the context of INRs. This limits the novelty of the paper. However, I am leaning towards acceptance since this reformulation is useful to the community in my opinion.

The proposed method (despite being efficient compared to a grid search) is expensive. Therefore it might not be useful in time consuming tasks such as NeRFs. The authors have also not included any NeRF experiments, which makes its applicability in one of the most popular tasks of INRs doubtful.

**Questions:**

Can you provide any metrics to showcase how efficient is the proposed method against a grid search?

Can you provide time durations taken for each task in finding the optimal configurations?

Can you provide any NeRF experiments with the proposed method?

---

> ### Author Response · Authors · 2025-11-25
> **To Reviewer 51xP**
>
> We sincerely thank the reviewer for their constructive feedback and supportive assessment, particularly for recognizing the usefulness of our reformulation to the INR community.
>
> ***C***: Comment; ***R***: Response
>
> ***C1***: Choosing correct hyperparameters for INRs is a cumbersome process which involves expensive grid searches...This limits the novelty of the paper. However, I am leaning towards acceptance since this reformulation is useful to the community, in my opinion.
>
> ***R1:*** We thank the reviewer for their supportive assessment and for highlighting the usefulness of our framework. We agree: our novelty lies not in the BO algorithm itself, but in the formalization of the problem it solves. Prior INR literature has relied on fragmented, ad-hoc heuristics to configure a complex, joint space of categorical activations (e.g., SIREN, WIRE) and their conditional continuous initializers. Our work is the first to formalize this unique INR design space as a single, global optimization problem - a space where we show greedy or layer-wise methods are not guaranteed to succeed. Thus, our contribution is to move the INR design from a manual art to a principled, automated, and globally aware process. We are grateful that the reviewer recognizes the value of this "reformulation" for the community.
>
> ***C2***: ...Therefore it might not be useful in time-consuming tasks such as NeRFs. The authors have also not included any NeRF experiments, which makes its applicability in one of the most popular tasks of INRs doubtful... Can you provide any NeRF experiments with the proposed method?
>
> ***R2:*** We appreciate the reviewer’s concern. To directly address the applicability of **OptiINR** to high-cost settings such as NeRF, we have **added new NeRF experiments** in the **revised PDF (Appendix D)**. The updated appendix includes **complete experimental setups, training details, and evaluation protocols**, showing that in **time-consuming tasks such as NeRF, OptiINR consistently outperforms other baselines**. In the final submission, we will further expand this section with **additional NeRF configurations and applications** to more comprehensively demonstrate that OptiINR remains both practical and effective for computationally intensive INR tasks.
>
> ***C3***: Can you provide any metrics to showcase how efficient is the proposed method against a grid search?
>
> ***R3:*** We thank the reviewer for this helpful suggestion. In response, we have added **both grid search and random search baselines** on **Kodak 24** in the **revised PDF (Appendix F, Table 6)** for direct comparison with OptiINR, including **evaluation counts, runtime, and best-achieved performance**. These results show that **OptiINR identifies superior configurations with substantially fewer evaluations**. We will incorporate these metrics into the revised manuscript and update the corresponding figures in the final submission.
>
> ***C4***: Can you provide time durations taken for each task in finding the optimal configurations?
>
> ***R4:*** We thank the reviewer for the suggestion. While OptiINR evaluates multiple configurations during BO, its purpose is to automate the search that practitioners already perform manually. To provide clarity, ***we include a brief discussion in Appendix F of the current revised submission.***

---

### Author Response · Authors · 2025-11-28
**Summary Comment by Authors**

Dear Area Chair,

We sincerely thank you and all reviewers for your time and constructive feedback. This work addresses a fundamental bottleneck in the INR literature: the reliance on ad-hoc heuristics for configuration. We propose **OptiINR**, the **first unified framework** that formulates INR design as a rigorous Bayesian optimization problem, automatically identifying optimal activation families and initialization scales. Our main rebuttal contributions include:

* **Expanded Scope (NeRF & PDEs):** We demonstrated OptiINR's effectiveness on complex inverse problems (NeRF) and scientific computing (PDEs), proving it extends beyond simple signal fitting.
* **Efficiency Analysis:** We provided concrete metrics showing OptiINR identifies superior configurations with substantially fewer evaluations than grid or random search.
* **Architectural Discovery:** We revealed that OptiINR automatically discovers principled architectural patterns (e.g., periodic $\to$ localized activations) that align with signal processing intuition but were previously found only through manual trial-and-error.
* **New Baselines:** We added comparisons against recent methods (SPDER, FreSh) to ensure comprehensive evaluation.

Reviewers recognized this as "**quite novel**" (Tceo), "**useful to the community**" (51xP), and noted that it will "**deepen the academic community's understanding**" of INRs (kaVS). Following our rebuttal and the addition of substantial new experiments, **Reviewer kaVS explicitly recommended acceptance**, and **Reviewer Vq9w maintained their positive rating**.

### 1. Major Revisions & New Experiments

**1.1 New NeRF and PDE Experiments (Addressing 51xP, Tceo, kaVS)**
A primary concern was the applicability of OptiINR to inverse problems and scientific tasks. We conducted extensive new experiments:
* **NeRF (Appendix D):** We applied OptiINR to volumetric rendering tasks. Results show OptiINR consistently outperforms baselines by discovering task-specific configurations (e.g., lower frequency scales for positional encoding, specific activation transitions).
* **PDE Solving (Appendix E):** We added a "Shear Flow" PDE benchmark. OptiINR successfully identified configurations that minimize solution error, validating the framework's utility in AI for Science.

**1.2 Efficiency vs. Baselines (Addressing 51xP, Tceo)**
We addressed concerns regarding computational cost by adding direct comparisons to Random Search and Grid Search in Appendix F.
* **Sample Efficiency:** OptiINR achieves higher PSNR with significantly fewer function evaluations compared to brute-force methods.
* **Cost Analysis:** We reported GFLOPs and parameter counts, confirming that OptiINR-optimized models incur no additional inference cost compared to standard INRs, as the base architecture remains identical.

### 2. Emerging Architectural Principles
**Addressing Reviewer kaVS’s request for "Architectural Patterns":**
We analyzed the global optima found by OptiINR to distill interpretable design principles (detailed in the revised manuscript):
* **Image Reconstruction:** OptiINR consistently selects periodic activations (Sine/FINER) in early layers to capture texture, switching to localized activations (WIRE) in deeper layers to refine details.
* **Audio:** Higher frequency scales and fully periodic architectures are preferred.
* **NeRF:** A transition from periodic inputs (for geometry) to smoother activations (ReLU/Gauss) for color emission density.

These findings validate that OptiINR moves beyond "black box" optimization to reveal **explainable design strategies**.

### 3. Conclusion
Following our rebuttal and supplementary experiments:
* **Reviewer kaVS** confirmed their concerns were resolved, highlighting the value of the discovered architectural patterns, and **recommended acceptance**.
* **Reviewer Vq9w** maintained a positive rating and requested a schematic workflow diagram, which we have committed to adding in the camera-ready version.
* **Reviewer 51xP** acknowledged the utility of the reformulation and leaned towards acceptance.
* **Reviewer Tceo** noted the novelty and contribution, with their specific concerns on compute and PDEs addressed in the appendices.

We respectfully request the AC to consider the positive consensus and the new empirical evidence demonstrating OptiINR's ability to automate and improve INR design across diverse modalities. We once again thank all reviewers for their constructive feedback and valuable time throughout this process.

Best regards,

Authors

---

### Note · Program_Chairs · 2026-01-17
**Submission Desk Rejected by Program Chairs**

The following references in this submission do not refer to real documents and/or have major errors in bibliographic information:

 Zheng, Y., Wang, Z., Zhang, D., and Wang, H. (2024). Fourier reparameterized training for implicit neural representations. In Proceedings of the IEEE/CVF Conference on Computer Vision and Pattern Recognition.